



# Quantifying $CO_2$ emissions of a city with the Copernicus Anthropogenic $CO_2$ Monitoring satellite mission

Gerrit Kuhlmann[1], Dominik Brunner[1], Grégoire Broquet[2], and Yasjka Meijer[3]

[1]Empa, Swiss Federal Laboratories for Materials Science and Technology, Dübendorf, Switzerland
[2]Laboratoire des Sciences du Climat et de l'Environnement, LSCE/IPSL, CEA-CNRS-UVSQ, Université Paris-Saclay, Gif-sur-Yvette, France
[3]European Space Agency (ESA), ESTEC, Noordwijk, The Netherlands

**Correspondence:** Gerrit Kuhlmann (gerrit.kuhlmann@empa.ch)

**Abstract.** We investigate the potential of the Copernicus Anthropogenic Carbon Dioxide ($CO_2$) Monitoring (CO2M) mission, a proposed constellation of $CO_2$ imaging satellites, to estimate the $CO_2$ emissions of a city on the example of Berlin, the capital of Germany. On average, Berlin emits about $20\,\mathrm{Mt}\,CO_2\,\mathrm{yr}^{-1}$ during satellite overpass (11:30 local time). The study uses synthetic satellite observations of a constellation of up to six satellites generated from one year of high-resolution atmo-

spheric transport simulations. The emissions were estimated by (1) an analytical atmospheric inversion applied to the plume of Berlin simulated by the same model that was used to generate the synthetic observations, and (2) a mass-balance approach that estimates the $CO_2$ flux through multiple cross-sections of the city plume detected by a plume detection algorithm. The plume was either detected from $CO_2$ observations alone or from additional nitrogen dioxide ($NO_2$) observations on the same platform. The two approaches span the range between the optimistic assumption of a perfect transport model that provides

an accurate prediction of plume location and $CO_2$ background, and the pessimistic assumption that plume location and background can only be determined reliably from the satellite observations. Often unfavorable meteorological conditions allowed to successfully apply the analytical inversion to only 11 out of 61 overpasses per satellite per year on average. From a single overpass, the instantaneous emissions of Berlin could be estimated with an average precision of 3.0 to $4.2\,\mathrm{Mt}\,\mathrm{yr}^{-1}$ (15-21% of emissions during overpass) depending on the assumed instrument noise ranging from 0.5 to 1.0 ppm. Applying the mass

balance approach required the detection of a sufficiently large plume, which on average was only possible on 3 overpasses per satellite per year when using $CO_2$ observations for plume detection. This number doubled to 6 estimates when the plumes were detected from $NO_2$ observations due to the better signal-to-noise ratio and lower sensitivity to clouds of the measurements. Compared to the analytical inversion, the mass balance approach had a lower precision ranging from 8.1 to $10.7\,\mathrm{Mt}\,\mathrm{yr}^{-1}$ (40-53%), because it is affected by additional uncertainties introduced by the estimation of the location of the plume, the $CO_2$

background field, and the wind speed within the plume. These uncertainties also resulted in systematic biases, especially without the $NO_2$ observations. An additional source of bias were non-separable fluxes from outside of Berlin. Annual emissions were estimated by fitting a low-order periodic spline to the individual estimates to account for the temporal variability of the emissions. The analytical inversion was able to estimate annual emissions with an accuracy of $<1.1\,\mathrm{Mt}\,\mathrm{yr}^{-1}$ (<6%) even with only one satellite. In contrast, at least two satellites were necessary for the mass-balance approach to have a sufficiently large

number of estimates distributed over the year to robustly fit a spline, but even then the accuracy was low ($>8\,\mathrm{Mt}\,\mathrm{yr}^{-1}$ (>40%))





when using the $CO_2$ observations alone. When using the $NO_2$ observations to detect the plume, the accuracy could be greatly improved to 22% and 13% with two and three satellites, respectively. Using the complementary information provided by the $CO_2$ and $NO_2$ observations on the CO2M mission, it should be possible to quantify annual emissions of a city like Berlin with an accuracy of about 10 to 20%, even in the pessimistic case that plume location and $CO_2$ background have to be determined

from the observations alone. This requires, however, that the temporal coverage of the constellation is sufficiently high to resolve the temporal variability of emissions.

## 1   Introduction

Anthropogenic carbon dioxide ($CO_2$) emissions will have to be reduced drastically in the coming decades to limit global

warming below the goals set in the Paris climate agreement (Rockström et al., 2017). Cities will play an essential role in solving this challenge, because they are responsible for over two-thirds of the global energy consumption and consequently for a large fraction of global $CO_2$ emissions (International Energy Agency, 2008). Recognizing their importance, many cities worldwide are now introducing stringent policies to reduce their carbon footprint and improve their resilience to climate change (e.g., C40 cities, 2018). However, tracking progress towards their reduction targets requires consistent, reliable, and timely

information on $CO_2$ emissions. Such information could be provided by atmospheric observations of the $CO_2$ concentrations over and downwind of cities, as demonstrated in a number of measurement campaigns such as the Indianapolis Flux Experiment (INLFUX) (Turnbull et al., 2018) or as part of the the Urban Climate Under Change [UC][2] project for the city of Berlin (Klausner et al., 2019). However, deducing emission fluxes from ground-based or airborne observations is not trivial and requires a large and expensive measurement infrastructure.

An alternative is to use satellite imaging spectrometers as already demonstrated for measurements of nitrogen dioxide ($NO_2$) over cities (Beirle et al., 2011; Lorente et al., 2019) and sulfur dioxide ($SO_2$) over large point sources (Fioletov et al., 2015). The advantage of satellite observations is that they measure the total amount of a gas in the vertical column rather than the concentration at a single point. Emissions can then be deduced from the divergence in the total column field (Beirle et al., 2019). The same concepts could be applied to $CO_2$, but this will require satellites with imaging capability similar to those available

for $NO_2$ and $SO_2$. The potential of such observations for quantifying $CO_2$ emissions has already been demonstrated in studies with synthetically generated observations for power plants (Bovensmann et al., 2010) and cities (Pillai et al., 2016; Broquet et al., 2018; Wang et al., 2020). The feasibility is further supported by recent studies using real $CO_2$ observations from the non-imaging Orbiting Carbon Observatory-2 (OCO-2) (Nassar et al., 2017; Reuter et al., 2019; Wu et al., 2020; Zheng et al., 2020).

Based on recommendations of a group of experts, which investigated the requirements of a future observing system to monitor anthropogenic $CO_2$ emissions (Ciais et al., 2015; Pinty et al., 2018; Janssens-Maenhout et al., 2020), the European

(c) Author(s) 2020. CC BY 4.0 License.





Commission and the European Space Agency (ESA) are currently preparing the Copernicus Anthropogenic $CO_2$ Monitoring Mission (CO2M), a constellation of polar orbiting $CO_2$ satellites with imaging capability (Sierk et al., 2019). According to the current system concept, the satellites will carry additional instruments with supporting observations of $NO_2$, aerosols and clouds. One prime goal of CO2M will be to support the quantification of emissions from hot spots including cities and power

plants.

The present study was carried out within in the framework of an ESA-funded project on the use of satellite measurements of auxiliary reactive trace gases for fossil fuel carbon dioxide emission estimation (SMARTCARB), for which Observing System Simulation Experiments (OSSEs) were conducted to provide guidance for the dimensioning of the CO2M mission and its instruments, in particular to assess the potential benefit of additional $NO_2$ measurements on the same platform (Kuhlmann

et al., 2019b). The OSSEs were based on high-resolution $CO_2$ and $NO_2$ simulations with the COSMO-GHG atmospheric transport model and on synthetic satellite observations generated from these simulations. Similar simulations were conducted in previous studies (Pillai et al., 2016; Broquet et al., 2018), but they did not have comparable spatial resolution, temporal coverage or detailed treatment of emissions and fluxes.

By driving the model with state-of-the-art high-resolution anthropogenic $CO_2$ emissions and biospheric $CO_2$ fluxes, the

synthetic observations should mimic true observations as closely as possible. In a companion paper, Brunner et al. (2019) demonstrated the importance of releasing anthropogenic emissions using realistic vertical profiles in atmospheric $CO_2$ simulations, because a large proportion of these emissions occur through stacks, notably from power plants. The present study is based on the same simulations, where stack height and meteorology-dependent plume rise were explicitly accounted for not only for power plants surrounding Berlin but also for the larger point sources within the city.

In this paper, we investigate how well the individual satellites of the CO2M mission will be able to quantify emissions of the city of Berlin during single overpasses, and how well a constellation of satellites will be able to estimate annual mean emissions. The emissions were estimated with and without coincident observations of $NO_2$ with different assumptions about the precision of the $CO_2$ instrument. Two complementary approaches were used encompassing the range between optimistic and pessimistic assumptions regarding the capability of atmospheric transport models. The first approach is an analytical

atmospheric inversion that is applied to the simulated plume signature of the city provided by the same model used to generate the synthetic observations. This approach follows the general concept used in previous OSSEs studies (Pillai et al., 2016; Broquet et al., 2018). Since this approach assumes that the atmospheric transport is known perfectly, it does not account for the effect of model errors on the estimated emissions, in particular, the challenge to correctly simulate the location of the emissions plume, which was also not considered in previous studies. The method also assumes that the $CO_2$ background field

from anthropogenic and natural fluxes outside the city can be obtained appropriately from the simulations.

To present an alternative to these optimistic assumptions, a mass-balance approach is used here as a second approach, which estimates the flux of $CO_2$ through control surfaces perpendicular to the main flow within the emissions plume (e.g. Beirle et al., 2011; Krings et al., 2013). A plume detection algorithm is required to determine the location of the plume in the satellite image. The location of the plume can also be used to obtain the $CO_2$ background field from satellite observations in the surroundings

of the detected plume. The algorithm used for detection has been presented in a second companion paper (Kuhlmann et al.,





2019a), which showed that the number of detectable plumes is significantly increased if additional $NO_2$ measurements are available on the same platform. Except for an estimate of the mean flow speed within the plume, the mass-balance approach is entirely data-driven and does not require any additional model information. This makes it possible to determine how accurately the emission can be quantified without considering prior knowledge from a model.

## 5  2  Data

The input data for this study are synthetic satellite observations from high-resolution $CO_2$ and $NO_2$ simulations that were generated in the SMARTCARB project. The model setup and the satellite scenarios are summarized in the following and are described in detail by Brunner et al. (2019) and Kuhlmann et al. (2019a).

### 2.1  Model simulations

$CO_2$, carbon monoxide (CO) and nitrogen oxides ($NO_x = NO + NO_2$) fields were simulated with the COSMO-GHG model, which is a version of the non-hydrostatic regional weather prediction model COSMO (Baldauf et al., 2011) extended for the simulation of passive trace gases such as greenhouse gases (Liu et al., 2017). The simulations were conducted for a domain centered over the city of Berlin and covering a large number of power plants in Germany and neighbouring countries. The simulation spans the whole year 2015 with 1 km × 1 km spatial resolution. Initial and boundary conditions were provided by
the operational COSMO-7 analyses of MeteoSwiss for meteorology with 7 km horizontal resolution, by the Copernicus CAMS operational products for NO and $NO_2$ with 60 km resolution (Flemming et al., 2015), and by special high-resolution runs of ECMWF for CO and $CO_2$ with 15 km resolution (Agustí-Panareda et al., 2014). Anthropogenic emissions were taken from the TNO/MACC-3 inventory (7 km × 7 km resolution) (Kuenen et al., 2014, for Version 2) and were merged with a detailed inventory for Berlin provided by the city authorities (AVISO GmbH and IE Leipzig, 2016). The emissions were vertically
distributed according to predefined vertical profiles per source category. For large point sources, plume rise was computed explicitly to account for the varying meteorological conditions (Brunner et al., 2019). Temporal variability was prescribed using fixed temporal profiles for hourly diurnal, weekly and seasonal variations per source category. Biospheric $CO_2$ fluxes were computed offline by the vegetation photosynthesis and respiration model (VPRM) at 1 × 1 km$^2$ spatial and hourly temporal resolution (Mahadevan et al., 2008).

The simulations included a total number of 50 different tracers of $CO_2$, CO and $NO_x$ that represented different sources and release altitudes, and included background tracers constrained at the lateral boundaries by the global-scale models. Two $CO_2$ tracers were included that represent biospheric $CO_2$ fluxes due to respiration and photosynthesis. To account for $NO_x$ chemistry in a simplified way, the $NO_x$ tracers slowly decay with an e-folding lifetime of 4 hours. $NO_x$ concentrations were converted to $NO_2$ concentrations offline using an empirical formula that is often used for representing $NO_x$-to-$NO_2$ ratios
downstream of emission sources (Düring et al., 2011).

Only a small number of these tracers were used in the present study. We used two $CO_2$ and two $NO_2$ tracers representing time-constant and time-varying emissions of Berlin, respectively. Furthermore, we created background tracers that contain





$CO_2$ or $NO_2$ fields from all emissions and biospheric fluxes as well as inflow from lateral boundaries except for the emissions of Berlin.

## 2.2 Synthetic satellite observations

The CO2M mission is a proposed constellation of polar orbiting satellites with Equator crossing times around 11:30 local time (Sierk et al., 2019). The main payload will be an imaging spectrometer for retrieving $CO_2$ from measurements in the near-infrared and in two shortwave infrared spectral channels. The current system concept envisages a pixel size of $4\,km^2$ and a swath width of at least 250-km. CO2M will also provide additional measurements of $NO_2$, aerosols, and clouds.

Synthetic satellite observations of column-averaged dry air mole fractions of $CO_2$ ($XCO_2$) and $NO_2$ tropospheric columns were generated for a hypothetical constellation of six CO2M satellites with $2 \times 2\,km^2$ spatial resolution and 250 km wide swaths. Each satellite has a sun-synchronous orbit with an overpass time of 11:30 local time (LT) and a repeat cycle of 11 days. The individual satellites are distinguished by their equator starting longitude for the first orbit, which were chosen such that the satellites are spaced with equal angular distance in a common orbit. The constellation of six satellites has therefore angular distances of 60 degrees. The individual satellites are designated by the letters a to f.

With a constellation of six satellites Berlin could be observed every day. For more realistic scenarios with fewer satellites, the constellation was divided into constellations of 1, 2 or 3 satellites (still with equal angular distances). This allows investigating the impact of the size of the constellation on the accuracy of the estimated $CO_2$ emissions.

The error characteristics of the $CO_2$ and $NO_2$ measurements were specified in the SMARTCARB project in close collaboration with ESA (Kuhlmann et al., 2019a). For $XCO_2$, three uncertainty scenarios were prepared with 0.5, 0.7 and 1.0 ppm random noise for a ground pixel with a vegetation surface and a solar zenith angle (SZA) of $50°$ (VEG50 scenario). The random errors were calculated based on solar zenith angle and surface reflectances using the error parametrization formula of Buchwitz et al. (2013). Amplifications of the random errors in the presence of cirrus clouds and aerosols as well as the influence of systematic errors on the $XCO_2$ measurements were not considered in our study. For $NO_2$ columns, we only used the high-noise scenario with a reference noise $\sigma_{ref}$ of $2 \times 10^{15}\,cm^{-2}$ or 20%, whatever was larger. The $NO_2$ noise was further modified based on cloud fraction roughly doubling the noise at 30% cloud fraction. Table 1 summarizes the uncertainty scenarios.

The synthetic observations were flagged as cloudy using the total cloud fractions simulated with the COSMO-GHG model. Since the $CO_2$ retrieval requires strict cloud filtering, we removed all pixels with cloud fractions larger than 1%. $NO_2$ retrievals can tolerate higher cloud fractions. We used a cloud threshold of 30% to flag cloudy pixels as often applied in satellite $NO_2$ studies (e.g., Boersma et al., 2011).





**Table 1.** Uncertainty scenarios and cloud flagging threshold for the instruments on-board the CO2M satellites. For the $NO_2$ measurements, either absolute or relative noise is used depending on which is larger.

| Scenario name | Species | Absolute noise | Relative noise | Cloud flagging |
|---|---|---|---|---|
| $CO_2$ low noise | $CO_2$ | 0.5 ppm | - | >1% |
| $CO_2$ medium noise | $CO_2$ | 0.7 ppm | - | >1% |
| $CO_2$ high noise | $CO_2$ | 1.0 ppm | - | >1% |
| $NO_2$ low noise | $NO_2$ | $1\times10^{15}$ molec. cm$^{-2}$ | 15% | >30% |
| $NO_2$ high noise | $NO_2$ | $2\times10^{15}$ molec. cm$^{-2}$ | 20% | >30% |

## 3  Methods

### 3.1  Analytical inversion applied to the simulated plume

The analytical inversion uses the $CO_2$ tracer representing the anthropogenic emissions of Berlin as simulated by the COSMO-GHG model. The method thus assumes perfect knowledge of atmospheric transport, which allows isolating the uncertainties

in the flux inversion due to instrument noise. The inversion uses a forward model that computes the vector $\boldsymbol{y}_{\mathrm{mod}}$ of size $m$ of model simulated values at the locations of all $XCO_2$ measurements within the plume. The plume was defined as those pixels for which the enhancement of the tracer is larger than a typical variability of the background field set to 0.05 ppm. The vector $\boldsymbol{y}_{\mathrm{mod}}$ is given by the equation

$$\boldsymbol{y}_{\mathrm{mod}} = \mathbf{H}x + \boldsymbol{y}_{\mathrm{BG}} \tag{1}$$

where $x$ is a scalar representing the $CO_2$ emission strength of Berlin. $\mathbf{H}$ is the observation operator representing the sensitivity of the $XCO_2$ signal to emissions. It was obtained from the $CO_2$ tracer simulated with constant emissions of Berlin. $\boldsymbol{y}_{\mathrm{BG}}$ is the $XCO_2$ background, which was computed from the model simulated fields excluding the emissions from Berlin, consistent with the assumption of a perfect model with accurately known transport and anthropogenic and biospheric fluxes outside of Berlin.

The emission of Berlin was found as maximum likelihood optimal estimate by minimizing the following cost function

$$\chi^2(x) = (\boldsymbol{y}_{obs} - \mathbf{H}x - \boldsymbol{y}_{\mathrm{BG}})^T \boldsymbol{S}_\varepsilon^{-1} (\boldsymbol{y}_{obs} - \mathbf{H}x - \boldsymbol{y}_{\mathrm{BG}}) \tag{2}$$

where $\boldsymbol{y}_{obs}$ is the measurement vector containing the synthetic $XCO_2$ observations. $S_\varepsilon$ is the error covariance matrix of the model-observation mismatch, which in our case of a perfect transport model corresponds to the measurement error covariance matrix. The diagonal elements of the error covariance matrix were set to the square of the absolute noise specified in Table 1.

The analytical inversion was applied to synthetic satellite observations with constant and time-varying emissions of Berlin. The uncertainty of the estimated emission was taken from the covariance matrix estimated by the least square fit. As a second measure of uncertainty, we computed mean bias (MB) and standard deviation (SD) of the differences between estimated and



true emissions. Thereby, the true emission was taken at 10:30 UTC during satellite overpass. The plume may also contain emissions emitted earlier in the day, but this information is not available from the model. The variation in the diurnal cycle of emissions is rather small in the hours prior to the satellite overpass (Figure S1 in supplement). Relative errors were computed relative to the annual mean at overpass time which is $16.9\,\mathrm{Mt\,CO_2\,yr^{-1}}$ for constant and $20.0\,\mathrm{Mt\,CO_2\,yr^{-1}}$ for time-varying

emissions. The latter is higher because emissions at 10:30 UTC are larger than daily mean emissions.

## 3.2  Mass-balance approach applied to the detected plume

The mass-balance approach estimates $CO_2$ emissions from the plume detected by a plume detection algorithm. It calculates the fluxes through vertical control surfaces that intersect the city plume after removal of the background. Under the assumption of steady-state conditions, this flux is equivalent to the emissions.

The plume detection algorithm described in Kuhlmann et al. (2019a) was applied to determine the position of the $CO_2$ plume in the satellite observations. The algorithm was applied either to the $XCO_2$ observations or to the auxiliary $NO_2$ observations. As shown in Kuhlmann et al. (2019a), the $NO_2$ plumes largely overlap with the $XCO_2$ plumes despite the fact that $NO_x$ is released in the model primarily at the surface by traffic emissions, whereas a larger proportion of $CO_2$ is released from stacks at higher altitudes. The number and size of the detected plumes was significantly larger when the algorithm was applied to the

$NO_2$ measurements due to their better signal-to-noise ratio and lower sensitivity to clouds.

The mass-balance approach computes $CO_2$ emissions from the total $CO_2$ mass flux through a vertical surface by integrating the satellite observations perpendicular to the direction of propagation of the plume:

$$M_p(x_p) = \int\limits_{y_{min}}^{y_{max}} c_p(x_p, y_p) dy_p \tag{3}$$

where $x_p$ and $y_p$ are along- and across-plume coordinates, $M_p$ is the line density with units of $\mathrm{kg\,m^{-1}}$, i.e. mass of $CO_2$ within

a slice of thickness of 1 m, and $c_p$ is the plume signal in $\mathrm{kg\,m^{-2}}$ converted from ppm. To obtain the total mass flux $F_p$, the line density is multiplied with the wind speed $u$ perpendicular to the control surface.

To obtain the plume signal $c_p$, the background needs to be subtracted from the satellite observations. The $CO_2$ background was estimated from the pixels surrounding the plume assuming that it is spatially smooth. To compute the background, all pixels within the detected plume were masked and replaced by interpolated values obtained by normalized convolution applied

to the unmasked pixels surrounding the plume. The normalized convolution was performed with a Gaussian filter with $\sigma = 10$ pixels, i.e. a width of the Gaussian kernel of about 20 km.

To obtain the location of the control surfaces, the center line of the plume was computed by fitting a two-dimensional curve to pixels within an extended plume area, which consisted of the detected plume as well as pixels within 50 km distance of the plume or the source. The surrounding pixels help stabilize the fit at the beginning and at the end of the detected plume. For

the curve fit, pixels were weighted with the local mean values above background calculated by the plume detection algorithm. Outside the detected plumes, pixels were weighted either with a small value of 0.05 ppm or $0.2 \times 10^{15}$ molec. $\mathrm{cm}^{-2}$ depending



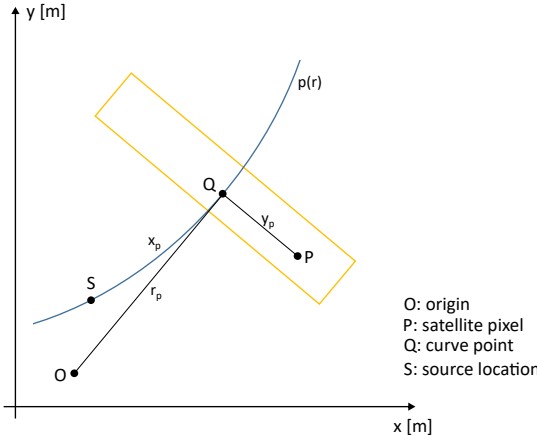

**Figure 1.** Sketch of the center line of a plume $p(r)$ with origin $\boldsymbol{O} = (x_o, y_o)$, satellite pixel $\boldsymbol{P}$, curve point $\boldsymbol{Q}$ and source location $\boldsymbol{S}$. The across-plume coordinate $y_p$ is the distance between $\boldsymbol{Q}$ and $\boldsymbol{P}$, and the along-plume coordinate $x_p$ is the arc length from $\boldsymbol{S}$ to $\boldsymbol{Q}$. The yellow rectangular is the polygon used for computing the line densities.

on whether $CO_2$ or $NO_2$ was used for plume detection. The two-dimensional curve $\boldsymbol{p}(r)$ consists of two parabolic polynomials:

$$p_x(r) = a_0 r^2 + a_1 r + a_2 \tag{4}$$

$$p_y(r) = b_0 r^2 + b_1 r + b_2 \tag{5}$$

with coefficients $a_k$ and $b_k$ and radial distance $r$. The parameter $r$ is calculated as the distance from the origin

$$5 \quad r = \sqrt{(x - x_o)^2 + (y - y_o)^2} \tag{6}$$

where $x$ and $y$ are easting and northing in the "DHDN / Soldner Berlin" spatial reference system (EPSG: 3068). The origin $(x_o, y_o)$ is placed at least 50 km away from the source in the direction opposite to the plume, i.e. in the west of Berlin if the plume is in the east and vice versa. Figure 1 shows an example for a detected plume with the origin $\boldsymbol{O}$ and source location $\boldsymbol{S}$.

To calculate line densities, pixel coordinates were converted to along- and across-plume coordinates for each pixel. The across-plume coordinate $y_p$ is the distance between pixel $\boldsymbol{P}$ and the curve at radial distance $r_p$, i.e. point $\boldsymbol{Q}$ in Fig. 1, for which the line from $\boldsymbol{Q}$ to $\boldsymbol{P}$ is perpendicular to the curve. The along-plume coordinate $x_p$ is the arc length of the center curve from the source origin $\boldsymbol{S}$ to $\boldsymbol{Q}$. $x_p$ and $y_p$ were calculated with a computationally efficient analytical solution as presented in the Supplement.

To obtain the control surfaces, we draw 10 km wide boxes (nearly rectangular polygons), perpendicular to the center line every 10 km along the plume. Choosing a box rather than a single line across the plume reduces the impact of noise and data gaps due to the larger number of available pixels. The across-plume width of the polygons is given by the maximum width of the detected plume plus an additional boundary of at least 10 km on each side to ensure that the entire plume is within the polygon even if only a part of the plume was detected.





Line densities were computed for each polygon by integrating the $XCO_2$ signal above background in the polygon. We tested two options for computing the integrals: (1) Integrating in across-plume direction $y_p$ by adding up the plume signals $c_p$ of all pixels whose center point is within the polygon. (2) Fitting a Gaussian function to the plume signals in across-plume direction and computing its integral. The first method does not make any assumption about the shape of the cross section, which is

an advantage for city plumes that can be quite complex. The disadvantage is that it is more difficult to deal with missing pixels, which lead to an underestimation of line densities if not properly accounted for. To solve this issue, we sub-divided the polygons in across-plume direction in 5-km wide sub-polygons, for which we computed a mean value from the available pixels. Finally, we integrated over the mean values of the sub-polygons. Polygons were not used if the mean values of at least one of the sub-polygons with detected plume pixels could not be computed due to missing values. Note that this criterion rejects

more line densities from plumes detected from the $NO_2$ observations than from the $CO_2$ observations, because the latter detect narrower plumes.

The second method has been used, for example, by Reuter et al. (2019). Fitting a Gaussian curve has the advantage that it automatically interpolates missing values. The disadvantage is that the transects of a city plume do not necessarily resemble a Gaussian curve. The Gaussian curve can be written as:

$$c_p(y) = \frac{q}{\sqrt{2\pi}\sigma} \exp\left(-\frac{(y-\mu)^2}{2\sigma^2}\right) \tag{7}$$

with line density $q$, shift $\mu$ and standard deviation $\sigma$. To avoid misfits, especially when pixels near the center of the plume are missing, a Gaussian was only fitted if at least one valid observation was available in each sub-polygon in the transect. When $NO_2$ observations are available, it is also possible to simultaneously fit a Gaussian curve to the $NO_2$ observations using the same standard deviation $\sigma$ for both the $CO_2$ and $NO_2$ curve. Since the $NO_2$ observations have a higher signal-to-noise ratio,

the width of the $CO_2$ curve is constrained by the $NO_2$ observations. This method was demonstrated by Reuter et al. (2019) and is used here as a third method.

An estimate of the mean flow speed within each plume transect is required to convert line densities to emissions. The mean flow speed is the projection of the $CO_2$-weighted wind vector onto the along-plume direction. Because we assume not to know the vertical and horizontal distribution of winds and $CO_2$ concentrations sufficiently well from a model, we take the average

wind speed at the location of Berlin in the lowest 500 m above surface assuming a well mixed boundary layer as a rough estimate. The wind profile was taken from the COSMO-GHG model simulations at satellite overpass time but could be taken from any meteorological analysis data. Not taking the wind speeds directly at the locations of the cross-sections was an attempt to account for uncertainties in the simulated winds that would be encountered with real rather than synthetic observations. To estimate the uncertainty in this simplified estimate of the mean flow speed, we also computed an effective wind speed for each

polygon by weighting the wind speeds with the model tracer containing the city plume. The effective wind speeds are then averaged downstream of Berlin to obtain a wind speed for each overpass.

Finally, the fluxes in each polygon, computed as the product of line density and wind speed, were averaged to obtain an estimate of the mean source strength. We only considered values more than 10 km downstream of the city center to ensure that





fluxes are obtained from outside of the city area. Uncertainties in the mean source strength were computed from the standard error of the individual line densities as well as by comparing with the true emissions at overpass.

To better understand the individual error components, the differences between estimated and true emissions were further analyzed using the detailed information available from the simulation. For this purpose, the mass-balance approach was ad-
ditionally applied to the detected plume using the noise-free $XCO_2$ observations, the true $CO_2$ background, and the effective wind speed. By replacing the uncertain information obtained from the observations with the accurate information from the model, four different types of error were distinguished:

1. The *method error* is the difference between the true emissions at satellite overpass and the emissions computed using the model information. It represents the intrinsic uncertainties of the method that arise from simplified assumptions such as
constant emissions and wind speed, from the plume detection algorithm, and from the fitting of the center line.

2. The *retrieval error* represents the impact of the random noise in the $CO_2$ observations on the calculation of line densities.

3. The *background error* is caused by errors in the estimation of the background field and its impact on the computed plume signals.

4. The *wind error* is the error that occurs if the mean wind speed between 0 and 500 m above Berlin is taken instead of the
effective mean wind speed within the plume.

Note that these different error types are still strongly related to the size of detected plume and thus, in the case of the $CO_2$-based plume detection, to the instrument noise scenario. These errors were therefore calculated separately for the different noise scenarios.

## 3.3   Estimating annual emissions

To estimate annual emissions and their uncertainties, the temporal variability of emission has to be considered, which includes seasonal, diurnal, and weekend versus weekday variations. Without accounting for this variability, annual mean estimates derived from a small sample of satellite overpasses at a fixed time of the day may be significantly biased.

In this study, only the seasonal cycle was estimated using a Hermite spline with periodic boundary conditions (see Supplement). The periodic boundary conditions help constrain the cycle in winter months, where only few data points are available.
To properly fit the seasonal cycle, we used a spline with four equidistant knots. The annual emissions were then estimated by integrating over the seasonal cycle. The uncertainty of the annual emissions was estimated by error propagation from the precision of the $CO_2$ emission estimates at individual satellite overpasses. Since annual emissions estimated in this way are only representative of emissions a few hours before the satellite overpass, the estimated emissions were compared with the emissions at overpass time. Uncertainties in the ratio between emissions at overpass time and daily mean emissions are thus
not taken into account.





**Table 2.** Performance of the analytical inversion for individual satellite overpasses in terms of mean bias (MB) and standard deviations (SD) of the difference between estimated and true $CO_2$ emissions of Berlin. More plumes could have been used for emission quantification for scenarios with low noise (second value in column Number of plumes), but the statistics were computed only for those plumes that could be used with the high noise scenario (first value) for better comparability of the results.

| Emissions | $\sigma_{VEG50}$ (ppm) | Number of plumes | Mean bias (MB) | | Standard deviation (SD) | |
| --- | --- | --- | --- | --- | --- | --- |
| | | | Mt yr$^{-1}$ | % | Mt yr$^{-1}$ | % |
| Time-constant | 0.5 | 60 / 74 | 0.0 | 0.2 | 1.8 | 10.5 |
| | 0.7 | 60 / 70 | 0.1 | 0.3 | 2.5 | 14.7 |
| | 1.0 | 60 / 60 | 0.1 | 0.5 | 3.5 | 21.0 |
| Time-varying | 0.5 | 59 / 73 | -0.1 | -0.6 | 3.0 | 14.8 |
| | 0.7 | 59 / 66 | -0.1 | -0.3 | 3.4 | 17.0 |
| | 1.0 | 59 / 59 | 0.0 | 0.2 | 4.2 | 20.9 |

## 4 Results and discussions

### 4.1 CO$_2$ emissions estimated by analytical inversion

The analytical inversion was applied to all $CO_2$ plumes observed by the $CO_2$ satellites for constant and time-varying emissions and for the low, medium and high noise scenarios with $\sigma_{VEG50}$ of 0.5, 0.7 and 1.0 ppm, respectively. Figures 2a and c show
the time series of estimated $CO_2$ emissions for the medium noise scenario ($\sigma_{VEG50} = 0.7$ ppm) with a constellation of three satellites. $CO_2$ estimates with uncertainties larger than $10 \, \mathrm{Mt \, yr^{-1}}$, i.e. 50% of the mean emissions at satellite overpass time for time-varying emissions, were discarded to remove plumes with very weak $CO_2$ signals or with a small number of pixels.

    The boxplots (panels b and d) summarize the differences between estimated and true emissions for all plumes observed by the six satellites. A constellation of six satellites was able to estimate emissions successfully, i.e. with an uncertainty smaller
than $10 \, \mathrm{Mt \, yr^{-1}}$, for 60 to 74 overpasses for time-constant and 59 to 73 overpasses for time-varying emissions, depending on the noise scenario. The average number of successful estimates was 11 per satellite and year but with a large range from 5 to 17 because of varying cloud coverage and because some orbits cover Berlin less frequently than others. Table 2 shows mean bias (MB) and standard deviations (SD) of the differences between estimated and true emissions. To compute comparable statistics for each instrument scenario, the statistics were computed only for the 60 and 59 plumes, respectively, for which the
uncertainties were less than $10 \, \mathrm{Mt \, yr^{-1}}$ in all three noise scenarios.

    The constant emissions are generally well captured within the uncertainty range determined by the measurement noise. The uncertainties of the individual estimates vary strongly because the amplitudes and sizes of the plumes differ from case to case due to differences in wind speeds, cloud cover, and incomplete coverage of the plume by the swath. The MB is close to zero for all noise scenarios.

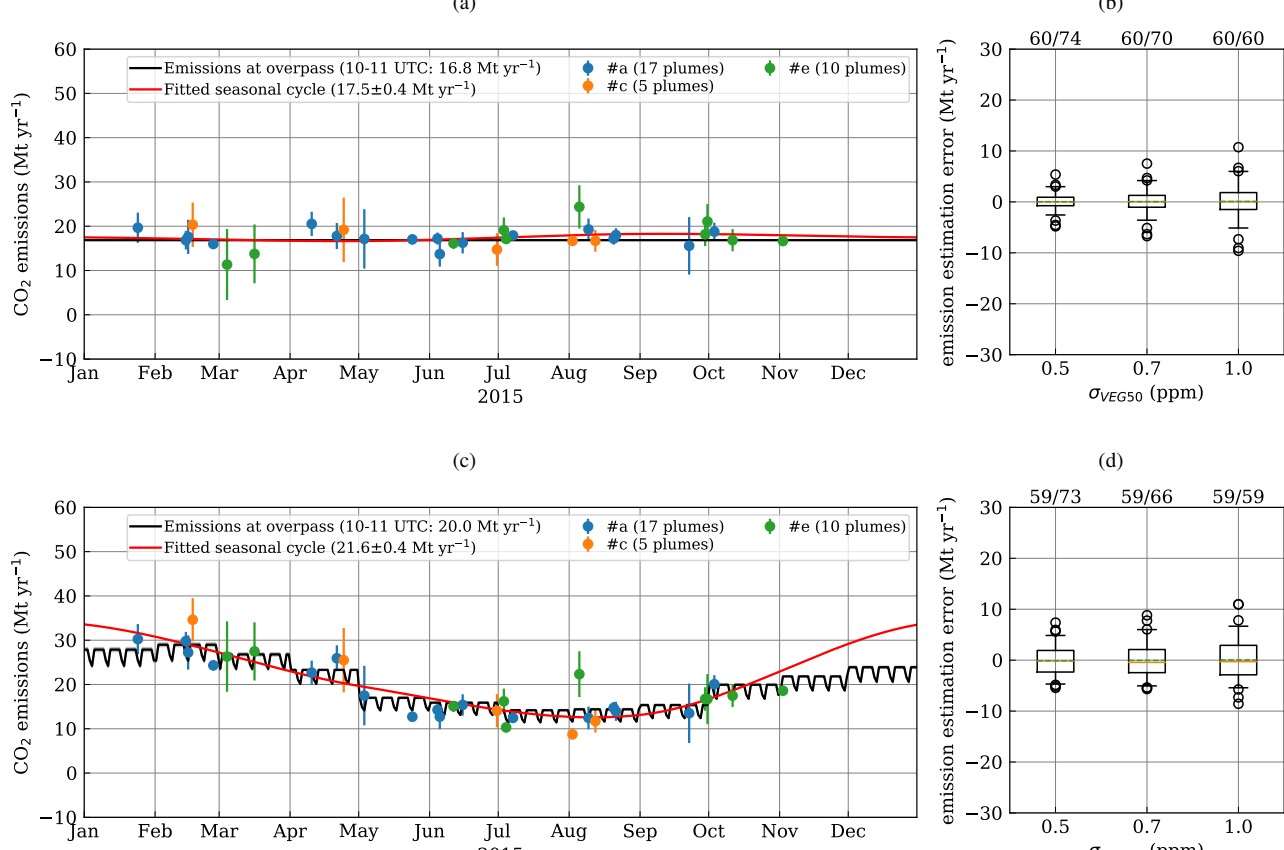

**Figure 2.** Time series of $CO_2$ emissions of Berlin estimated with the analytical inversion using three satellites (#a, #c and #e) with $\sigma_{VEG50}$ of 0.7 ppm for (a) constant and (c) time-varying emissions. Emission estimates with uncertainties larger than $10.0\,\mathrm{Mt\,yr^{-1}}$ (50% of mean emissions at 11:30 local time) were removed. (b,d) Boxplots of the difference between estimated and true $CO_2$ emissions of Berlin using six satellites for the three different instrument noise scenarios. The boxes denote the range between 25th and 75th percentiles, orange lines are median values, dashed lines are mean values, and whiskers are 5th and 95th percentiles. Only those 60 and 59 results were used for constant and time-varying emissions, respectively, for which the uncertainties are less than 50% for all three scenarios.

In case of time-varying emissions, the seasonal cycle of the emissions can be reproduced quite accurately because many plumes can be observed with three or more satellites and because the individual estimates have an average uncertainty of only 14-21% depending on instrument noise scenario. The rare opportunities for observing plumes in winter, however, can easily be missed by a small constellation of satellites, which will make it difficult to reliably trace the seasonal cycle. The MB slightly

5 deviates from zero (Table 2) mainly because the observation operator $\mathbf{H}$ was calculated assuming constant emissions, while the measurement vector contains observations of time-varying emissions from several hours before the satellite overpass time. The SD of the differences between the individual emission estimates and the true emissions are 3.0, 3.4 and $4.2\,\mathrm{Mt\,yr^{-1}}$ for





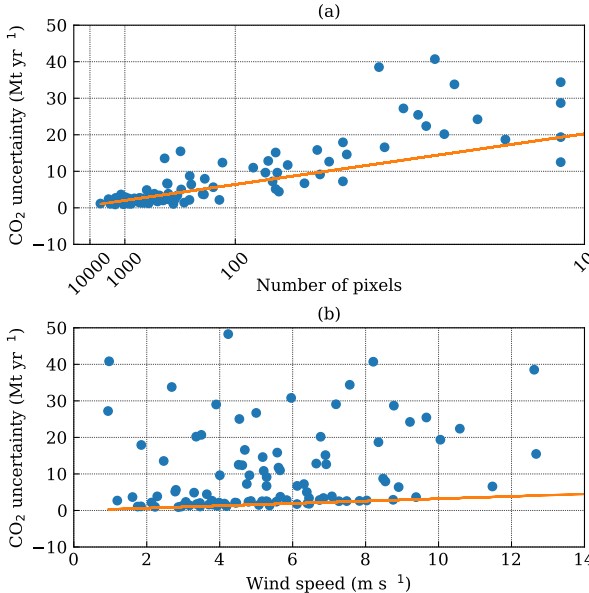

**Figure 3.** Dependency of the theoretical $CO_2$ uncertainty on (a) inverse of the square root of the number of pixels and (b) wind speed. A multilinear regression model was used to determine the slope of the linear dependence on these quantities.

$\sigma_{VEG50}$ of 0.5, 0.7 and 1.0 ppm, respectively (Table 2). These values agree well with the mean of the estimated uncertainties suggesting that the error propagation yields a realistic estimate of uncertainties.

The theoretical uncertainty computed by the inversion agrees well with the SDs computed for constant and time-varying emissions in Table 2. This estimated uncertainty depends on the number of pixels and the signal strength of the plume. The signal strength in turn depends on the wind speed and turbulent mixing. Figure 3 shows the dependency of the theoretical uncertainty on the inverse of the square root of the number of pixels and on wind speed for the medium noise scenario. Fitting a robust linear regression model yielded:

$$\sigma_{em} = \left[ (91.5 \pm 3.8) \frac{\sigma_{VEG50}}{\sqrt{n}} + (0.32 \pm 0.08)u \right] Mt\,yr^{-1} \tag{8}$$

with number of pixels $n$ and wind speed $u$. The uncertainty depends strongly on the number of pixels and is smaller than $10\,Mt\,yr^{-1}$ (50%) if the number of pixels is larger than 100 for all three noise scenarios. The dependency on wind speed is less robust and does not depend on the noise scenario. Most outliers are due to plumes of less than 100 pixels. Note that the fit coefficients are specific to the emissions and meteorological conditions of Berlin and can not be generalized for other cities.

## 4.2 $CO_2$ emissions estimated by mass balance approach

The mass-balance approach was applied to synthetic observations of the CO2M mission for a constellation of up to six satellites using the uncertainty scenarios with $\sigma_{VEG50}$ of 0.5, 0.7 and 1.0 ppm. The location of the $CO_2$ plumes was either detected from the $CO_2$ observations alone or from the additional $NO_2$ observations on-board the same satellite.

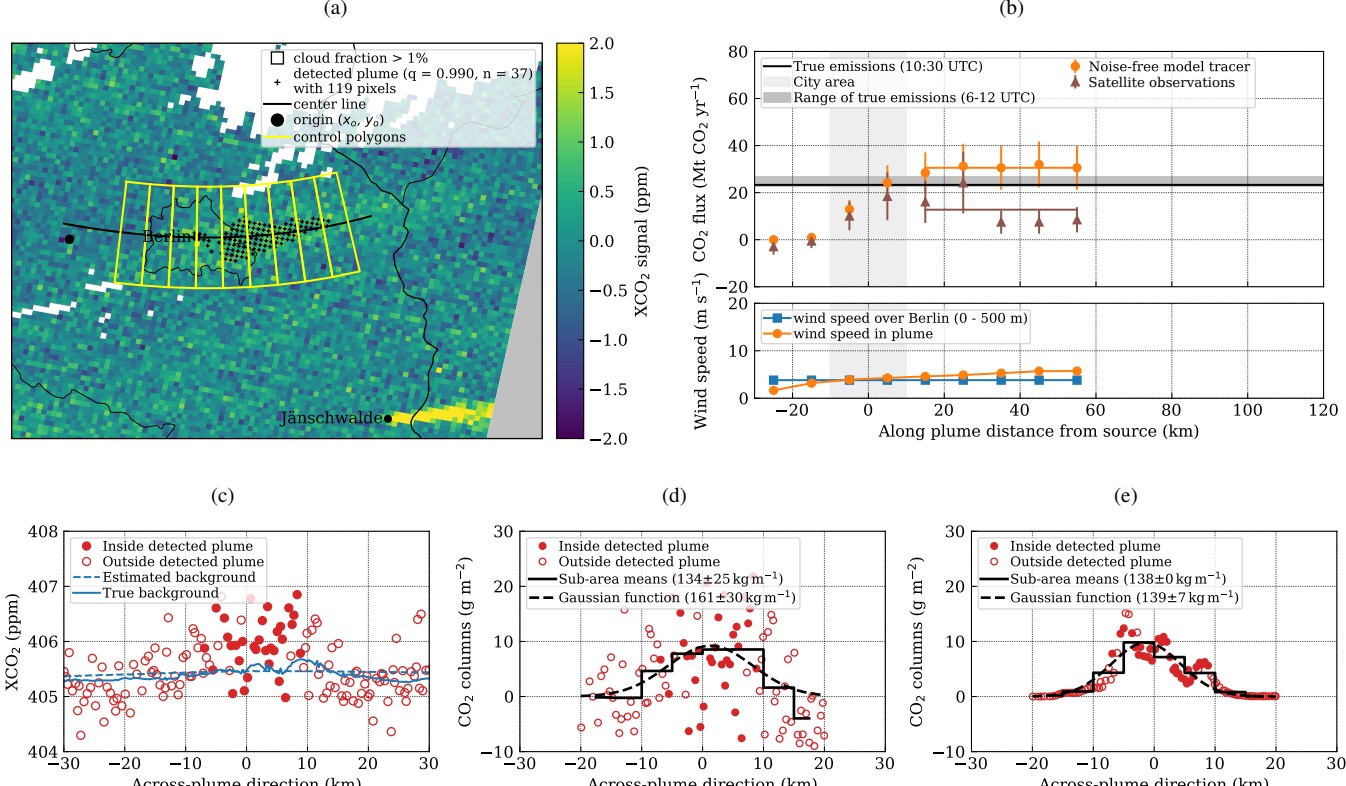

**Figure 4.** Illustration of the mass-balance approach applied to a plume on 23 April 2015 observed with low-noise $CO_2$ observations: (a) detected $XCO_2$ plume with polygons used for computing the line densities. (b) Line densities as a function of along-plume distance. (c) $XCO_2$ satellite observations as a function of across-plume distance in the polygon between 10 and 20 km downstream of the source. (d) same as (c) but $CO_2$ column densities after subtracting the estimated background field. Sub-polygon means and Gaussian fit are also shown. (e) $CO_2$ column densities from the model tracer containing only emissions of Berlin.

#### 4.2.1 Example for 23 April 2015

The method is illustrated in Figures 4 and 5 for a plume on 23 April 2015. The plume detected from the $CO_2$ observations (Fig. 4) is significantly smaller than the plume detected from the $NO_2$ observations (Fig. 5) (119 versus 780 pixels). The plume detected from the $CO_2$ observations is also shorter with a length of 60 km as opposed to 120 km. This results in having fewer polygons for computing line densities. Finally, the plume detected from the $CO_2$ observations is also narrower suggesting that a significant part of the real plume is attributed to the background.

The $CO_2$ concentrations were plotted in across-plume direction for the polygon between 10 and 20 km downstream of the centre of Berlin (Figures 4c-e and 5c-e) The $XCO_2$ signal in the plume is only about 1 ppm above background, which is comparable to the instrument noise of 0.5 to 1.0 ppm of the three instrument scenarios. A 1 ppm enhancement approximately

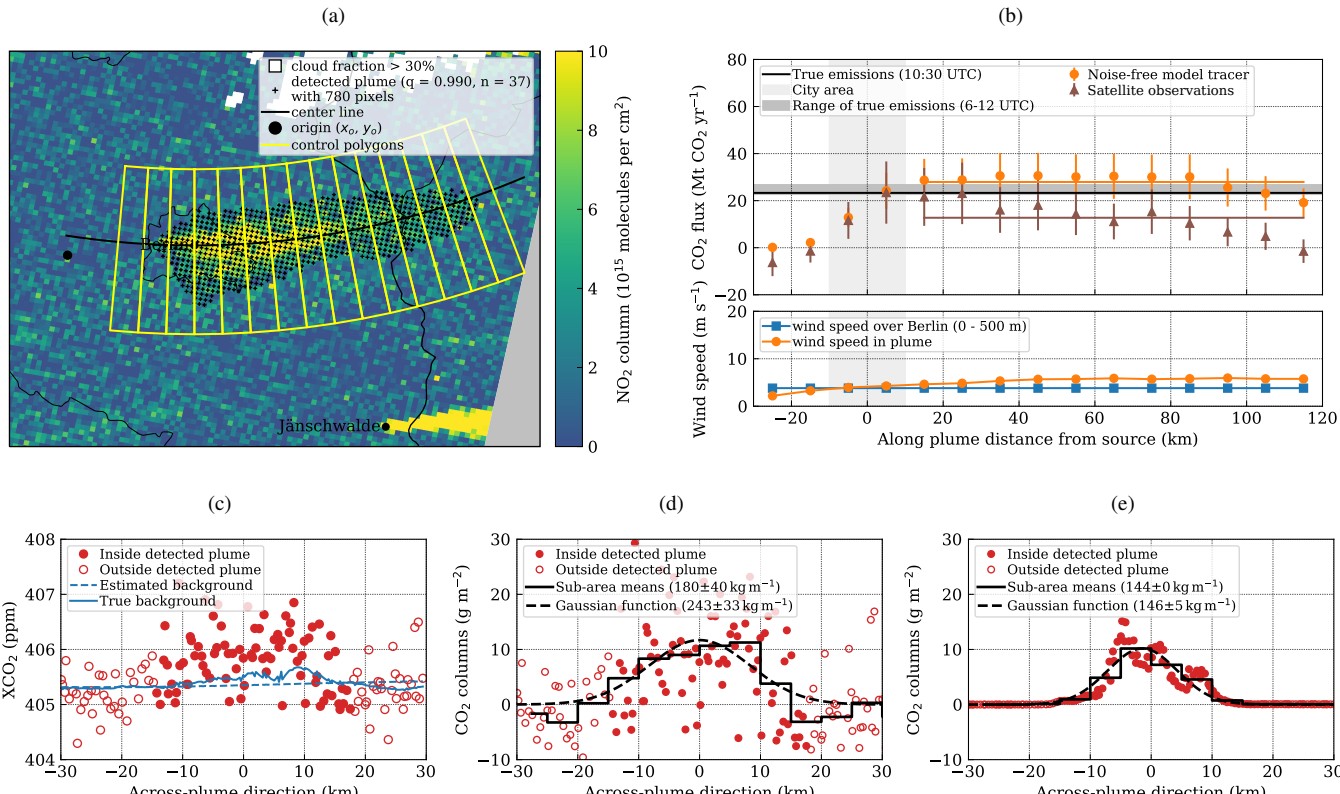

**Figure 5.** Same as Fig. 4 but using the $NO_2$ observations for detecting the plume.

corresponds to a $CO_2$ column density of $10\,g\,m^{-2}$. The model tracer representing the emissions of Berlin shows three distinct enhancements rather than a single Gaussian-shaped plume caused by the three power stations in Berlin (Fig. 4e). Nonetheless, the line density obtained by fitting the values from the model tracer with a Gaussian agrees well with the line density computed from the mean values in the sub-polygons. Note that the line densities were only computed between -20 and 20 km from the

5  center line, because pixels more than 10 km outside the detected plume were masked to avoid issues from neighboring plumes or variability in the $CO_2$ background. Since the plume detected from the $NO_2$ observations is wider, the line densities from the model tracer are slightly higher, because the plume edges still contain some $CO_2$ emitted from Berlin (Fig. 5e).

Figure 4d shows the across-plume column densities from the satellite observations after subtracting the estimated background. The line densities are $134\pm25\,kg\,m^{-1}$ and $161\pm30\,kg\,m^{-1}$ using the mean values in the sub-polygons and the Gaus-

10  sian fit, respectively. The uncertainty was computed from the random noise of the measurements for the sub-polygon means and from the quality of the fit for the Gaussian function. Figure 5d shows the same for the plume detected from the $NO_2$ observations. In this case, the line densities are higher with $180\pm40\,kg\,m^{-1}$ and $243\pm33\,kg\,m^{-1}$. The reason for these differences is on the one hand the larger and slightly shifted polygon due to the different plume detection and on the other hand due to the background error (see Section 4.3 for details).



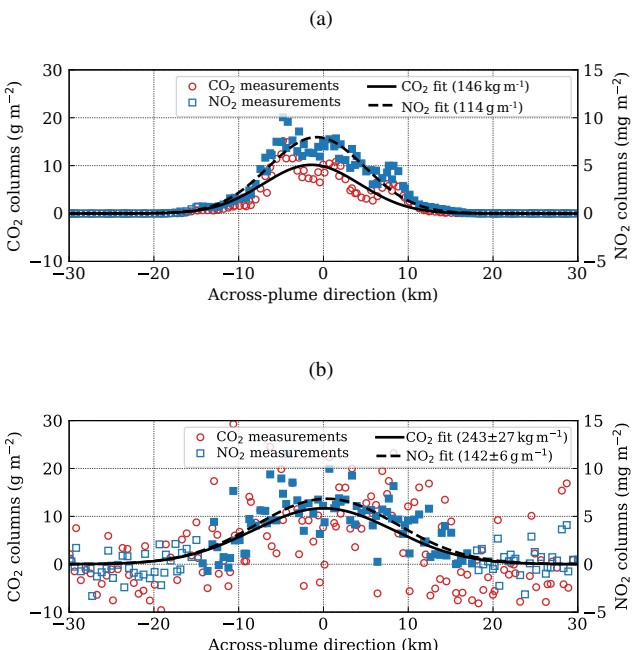

**Figure 6.** Line densities computed from (a) the model tracer and (b) synthetic satellite observations using the Gaussian curve constrained by the $NO_2$ observations.

The line densities in along-plume direction are shown in Figures 4b and 5b. Upstream of the city, the line densities are close to zero and then slowly build up over the city. They reach their maximum downstream of the city and stay constant because $CO_2$ does not decay in the atmosphere. The $CO_2$ fluxes computed from the noise-free model tracer are higher than the true emissions in this example, which is caused by the simplifications in the mass-balance approach, mainly by the assumption

of a constant flow parallel to the fitted center curve. The fluxes computed from the synthetic satellite observations are lower than the true emissions at overpass. Near the source, the error is quite small, but it gets larger downstream mainly due to growing systematic errors in the estimation of the background, since it gets increasingly difficult to separate the plume from the background in the fading plume.

Figure 6 shows the same $CO_2$ across-plume column densities from the model tracer and the satellite observations as Figure

5, but it shows additionally the $NO_2$ column densities. The $CO_2$ line density was obtained by fitting a Gaussian curve whose width was constrained by the additional $NO_2$ observations. While the line density is the same as without constraining the width of the curve for the present example, the estimated uncertainty is smaller.

### 4.2.2   Time series of estimated emissions

With six satellites, the $NO_2$-based plume detection could identify about 40 plumes suitable for applying the mass-balance

approach. On average, seven plumes were identified per satellite, but with a very large spread between the satellites (range: 1 -





**Table 3.** Mean bias (MB) and standard deviation (SD) of differences between estimated $CO_2$ emissions and emissions at overpass time (10-11 UTC) for Berlin based on observations with six satellites. The $CO_2$ plume was either detected from $CO_2$ or $NO_2$ observations (high noise scenario). The results are for line densities computed from the sub-polygon means.

| Plume detection | $\sigma_{\text{VEG50}}$ (ppm) | Number of plumes | Median plume size | Mean bias (MB) | | Standard deviation (SD) | |
| --- | --- | --- | --- | --- | --- | --- | --- |
| | | | | Mt yr$^{-1}$ | % | Mt yr$^{-1}$ | % |
| $CO_2$-based | 0.5 | 16 / 17 | 154 | 2.4 | 12.2 | 9.1 | 45.6 |
| | 0.7 | 16 / 16 | 137 | 1.1 | 5.3 | 8.1 | 40.3 |
| | 1.0 | 16 / 16 | 80 | 0.6 | 2.8 | 9.0 | 45.0 |
| $NO_2$-based | 0.5 | 34 / 34 | 654 | 2.6 | 13.0 | 10.1 | 50.6 |
| | 0.7 | 34 / 34 | 654 | 2.9 | 14.7 | 10.3 | 51.4 |
| | 1.0 | 34 / 34 | 654 | 3.5 | 17.4 | 10.7 | 53.3 |

14) because some orbits are more suitable than others for observing Berlin as mentioned earlier. In addition, since the number of overpasses is small, the uneven distribution of cloud-free days in time can have a large effect on the number of plume observations for a given satellite. From the $CO_2$ observations alone only about half of these plumes could be detected because of the lower signal-to-noise ratio and the more stringent cloud filtering required for the $CO_2$ observations.

Because of the different cloud thresholds for $CO_2$ and $NO_2$, some of the plumes detected from the $NO_2$ observations do not have enough cloud-free $CO_2$ pixels for computing line densities and can therefore not be used for estimating emissions. The $NO_2$-based plume detection generally results in significantly more pixels per plume with about 400 to 800 pixels compared to less than 300 pixels for $CO_2$-based detection. More details about the detectability of $CO_2$ plumes from the $CO_2$ and $NO_2$ observations are presented in Kuhlmann et al. (2019a).

Before applying the mass-balance approach, the detected plumes and the center lines were visually inspected to remove plumes with obvious issues. In particular, for the $CO_2$-based plume detection several plumes were removed for which the number of detected pixels was too small to reliably fit a center line parallel to the wind direction. In most cases, 50 or more detected pixels were sufficient. Often less than 100 pixels were detected from the high noise $CO_2$ observations, but it was often still possible to use these plumes with less than 100 pixels for estimating emissions. Three plumes were removed where the

$CO_2$ plume of the Jänschwalde power plant overlapped with the plume of Berlin. In many cases, the $CO_2$ image alone did not show clearly which plumes had a reasonable center line, and therefore additional information such as wind fields is helpful. The $NO_2$ images are also extremely helpful, because they are less affected by clouds and often reveal weak interfering plumes in the surrounding area that are not detectable from the $CO_2$ observations.

The number of plumes remaining for reliable emission estimation was 34 for plumes detected from the $NO_2$ observations,

which are only between 0 and 10 plumes per satellite and year with an average number of 5.7. These numbers would be approximately halved with the $CO_2$ observations alone: 16 to 17 plumes could be used with six satellites for the different noise scenarios, i.e. on average only 2.7 (range: 1-7) per satellite.





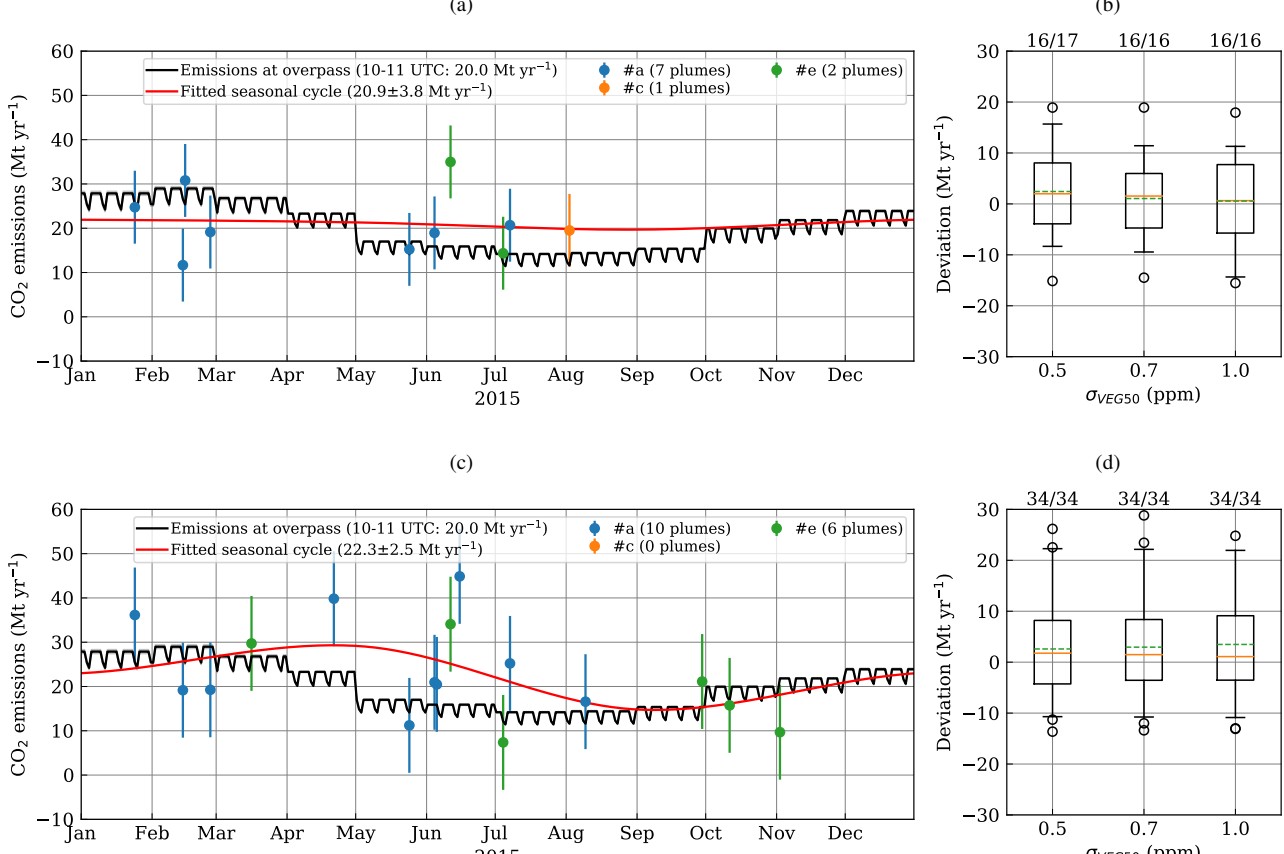

**Figure 7.** Time series of estimated $CO_2$ emissions of Berlin using a constellation of three satellites (#a, #c and #e) with medium noise instruments ($\sigma_{VEG50}$ = 0.7 ppm). The plumes were detected from (a) the $CO_2$ and (c) the $NO_2$ observations. The error bars show constant errors of $10.0\,\mathrm{Mt\,yr^{-1}}$ corresponding to the standard deviation of the differences between estimated and real emissions. (b, d) The boxplots show the difference between estimated and $CO_2$ emissions at overpass of Berlin for six satellites. The boxes denote the range between 25th and 75th percentiles, orange lines are median values, and whiskers are 5th and 95th percentiles.

Figures 7a and c present the time series of estimated $CO_2$ emissions for these plumes. The time series is shown for a constellation of three satellites for the medium noise scenario ($\sigma_{VEG50}$ = 0.7 ppm) and line densities computed from the sub-polygon means. The time series for a less likely constellation of six satellites are shown in the supplement. The error bars show constant errors of $10.0\,\mathrm{Mt\,yr^{-1}}$ corresponding to the SD of the differences between estimated and real emissions.

5   A constellation of three CO2M satellites without additional $NO_2$ observations can detect plumes and estimate emissions from only ten overpasses. These overpasses are likely to cluster in specific months with good weather conditions leaving significant gaps in other months. A constellation of six satellites is able to detect more plumes proving better temporal coverage





of the different seasons. This can also be achieved when $NO_2$ observations are available for plume detection. In this case, 16 plumes can already be used with a constellation of three satellites.

Note that for a few overpasses where the emissions were estimated from the plume detected from the $CO_2$ observations (e.g., the only estimate from satellite #c), the emissions were not estimated from the plume detected from the $NO_2$ observations. In these cases, only a fraction of the full plume was detected from the $CO_2$ observations, because the plume was partly covered by clouds, while a larger plume could be detected from the $NO_2$ observations. Since our algorithm rejects line densities with missing observations in sub-polygons, no emissions were calculated for cases where this affected all line densities.

## 4.3 Uncertainties in the mass-balance approach

As expected, uncertainties in the estimated emissions are larger for the mass-balance approach compared to the analytical inversion. Figure 7b and d show the difference between estimated emissions and the emissions at overpass for different noise levels of the $CO_2$ instrument. These box plots include all estimates for a constellation of six satellites.

Table 3 shows MB and SD of these differences. For better comparability, these statistics were computed only from those plumes that could be detected with all three noise scenarios, i.e. 16 plumes in case of $CO_2$-based detection and 34 plumes in case of $NO_2$-based detection. SDs are about $10\,\mathrm{Mt\,yr^{-1}}$, i.e. about 50% of the $20.0\,\mathrm{Mt\,yr^{-1}}$ emissions of Berlin at overpass time, which is about three times larger than for the analytical inversion. For the plume detection based on the $CO_2$ observations, SDs are about $9\,\mathrm{Mt\,yr^{-1}}$ (45%) and, interestingly, do not depend significantly on the noise level of the instrument. SDs are slightly larger with $10\,\mathrm{Mt\,yr^{-1}}$ (50%) if the $NO_2$ observations are used for plume detection, because applying the mass-balance approach to larger plumes is more challenging.

The MB is positive for both $CO_2$- and $NO_2$-based plume detection. With $NO_2$-based plume detection it rises slightly from 2.6 to $3.5\,\mathrm{Mt\,yr^{-1}}$ (13 - 17%) from the low to the high-noise scenario. With $CO_2$-based detection, in contrast, the lowest MB is surprisingly obtained for the high-noise scenario.

The MB is caused by systematic errors in the method, retrieval, background, and wind errors, which can have substantial systematic errors that may add up or compensate each other in the total error. The results therefore need to be interpreted with great care. MB and SD of the individual error components are summarized for the different noise scenarios in Figures 8 and 9 for $CO_2$ and $NO_2$-based plume detection, respectively. The method and total errors are computed against the true emissions at overpass, while the other errors are compared to the emissions computed using the model information. The total errors are identical to the errors presented in Table 3. The relative MB and SD are tabulated in the supplement. The different errors are discussed in detail in the following.

### 4.3.1 Method error

The mass-balance approach relies on assumptions and simplifications that result in uncertainties in the estimated emissions. The main sources of uncertainty are the assumption of constant emissions and constant flow parallel to the center line fitted to the detected plume. The method error also indirectly depends on the instrument noise scenario that affects the size of the detected plumes.




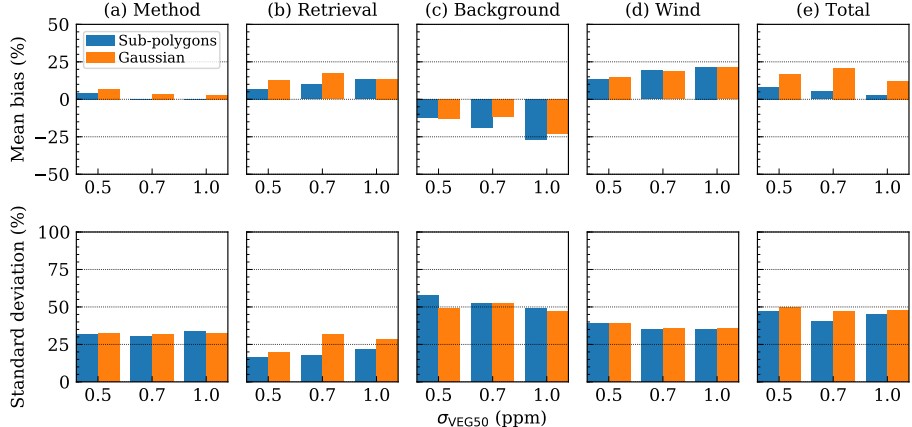

**Figure 8.** Mean bias (MB) and standard deviations (SD) of (a) method, (b) retrieval, (c) background, (d) wind and (e) total errors for $CO_2$-based plume detection. MB and SD are shown for the three uncertainty scenarios and for line densities computed from sub-polygon means and Gaussian function, respectively.

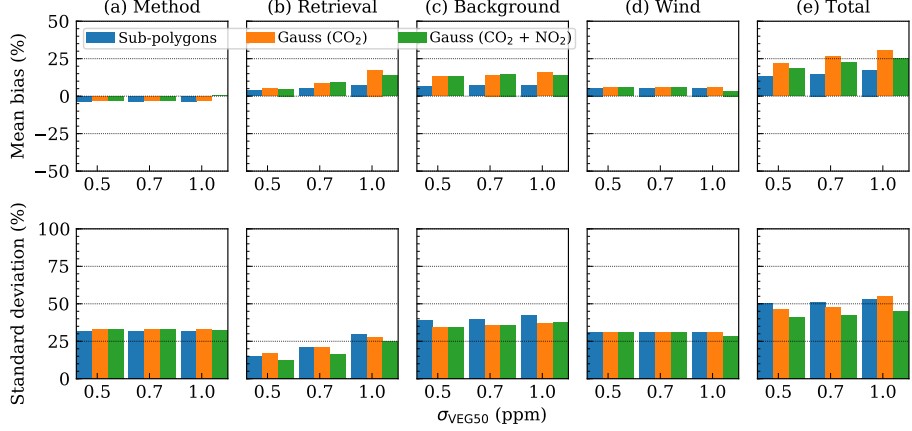

**Figure 9.** Same as Fig. 8 but for plumes detected from $NO_2$ observations.

Figures 8a and 9a show that the MB is slightly positive when plumes were detected from $CO_2$ observations and slightly negative when plumes were detected from $NO_2$ observations. The absolute MB is mostly smaller than 5% suggesting that the assumptions in the mass-balance approach do not cause significant systematic errors. In particular, we do not find that emissions are overestimated, although we compared estimated emissions with emissions at overpass (10:30 UTC), while the plume may also contain $CO_2$ released a few hours earlier when emissions are higher (Figure S1 in supplement). Note that such a bias would also affect the analytical inversion.





The SD of the method error is about 30%, which gives a rough estimate of the minimum error achievable with the mass-balance approach. The SD does not depend on the instrument noise scenario or whether $CO_2$ or $NO_2$ was used to detect the plume despite the influence this has on the size of the detected plume.

### 4.3.2 Retrieval error

The retrieval error shown here is affected mainly by the computation of the line densities from the noisy $CO_2$ observations, because the effect of the retrieval error on the plume detection algorithm is part of other error components. Figures 8b and 9b show that the retrieval error roughly doubles from the low noise scenario to the high noise scenario, which is consistent with the doubling of the random error in the instrument scenarios. The SD is similar to the uncertainty found in the analytical inversion (Table 2), which also only includes errors from the instrument noise.

The retrieval error has a small positive bias that is about twice the expected standard error of the sample computed from the SD and the number of estimates (i.e. $SD/\sqrt{n}$). However, the MB is still within two SDs and thus not too unlikely considering the relatively small number of samples. The MB scales with the absolute noise of the uncertainty scenarios, which is explainable by the fact that the same random errors, i.e. spatial noise pattern, were applied to the three uncertainty scenarios of a scene except for a scaling required to achieve the standard deviation of the error $\sigma_{\mathrm{VEG50}}$. This was done because the plume detection

algorithm is very sensitive to the noise pattern especially when using the $CO_2$ observations (Kuhlmann et al., 2019a). Applying different noise pattern would add an additional source of statistical fluctuation that would make it more difficult to compare the results of the different scenarios.

  If line densities are calculated by fitting a Gaussian function to the plume detected from the $CO_2$ observations, SDs are higher likely due to the challenge of fitting the curve through data with a low signal-to-noise ratio and because the detected

plume is narrow and does not include many background values outside the plume that would help stabilize the baseline (Figure 4d). Furthermore, the transect of the city plume often does not resemble a Gaussian curve, which results in an additional fitting error. In contrast, SDs are reduced when the curve is fitted by constraining its width using the $NO_2$ observations resulting in the lowest retrieval errors.

### 4.3.3 Background error

The $CO_2$ background has a strong impact on the estimated emissions, because a bias in the background field results in a bias in the $CO_2$ signals and thus in the estimated emissions. Figure 10 presents two examples of estimated and true $CO_2$ backgrounds for 27 February and 23 April 2015, respectively. The true $CO_2$ background was taken from the model tracer that includes all emissions and fluxes except emissions from Berlin. The background fields in the mass balance approach were estimated using the plume detected from the $NO_2$ observations (black dots). On 27 February, the $CO_2$ background field has a strong horizontal

gradient and the wind speed is relative low with $2\,\mathrm{m\,s^{-1}}$. On 23 April, the background has no gradient and the wind speed is somewhat higher with $6\,\mathrm{m\,s^{-1}}$. In general, the estimated background field is smoother than the true background field, which displays fine scale patterns associated with meteorology and $CO_2$ fluxes. The MB of the differences within the detected plume is -0.03 ppm on 27 February and +0.04 ppm on 23 April, which is small compared to the amplitude of Berlin's plume signal

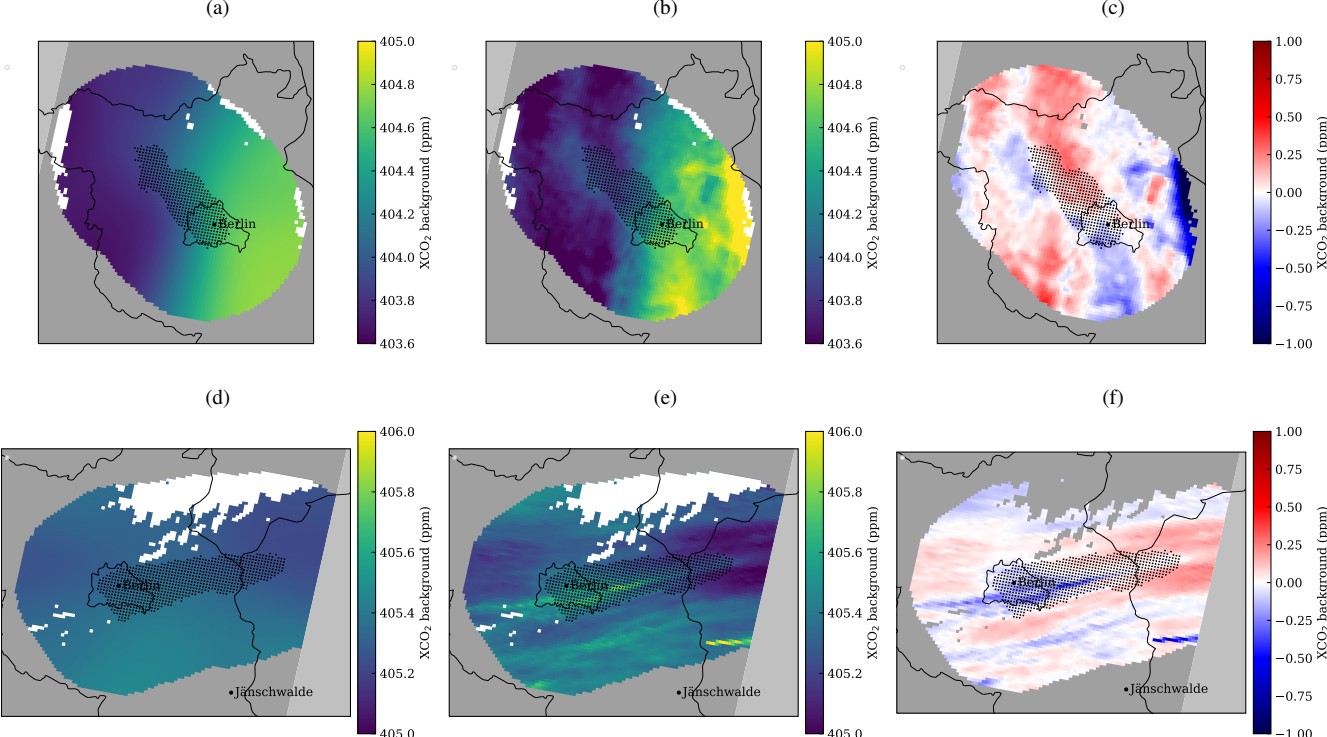

**Figure 10.** (a) Estimated background, (b) true background and (c) difference between estimated and true background for 27 February 2015. Black dots show the $CO_2$ plume detected from the $NO_2$ observations. Mean bias (MB) and standard deviation (SD) within the detected plume are -0.03 ppm and 0.11 ppm. (d,e,f) The same for the 23 April 2015 with MB and SD of 0.04 ppm and 0.11 ppm, respectively. Note that MB and SD are much smaller than the noise of the measurements. The effective wind speeds within the plumes are $2\,\mathrm{m\,s^{-1}}$ and $6\,\mathrm{m\,s^{-1}}$, respectively.

of about 1 ppm. The SD is 0.11 ppm within the detected plume for both overpasses, which is much smaller than the noise of the $CO_2$ observations ($\sigma_{\mathrm{VEG50}}$: 0.5 ppm to 1.0 ppm).

However, the differences between estimated and true background vary spatially with local biases up to $\pm 0.3$ ppm (see panels c and f). The size, shape and orientation of these patches depend on wind speed and direction. The patterns are caused by

5    the effect of meteorology on $CO_2$ from anthropogenic emissions outside of Berlin and biospheric fluxes inside and outside of Berlin. Since the size of the patches is similar to the size of the polygons used for computing the line densities, the local biases in the background can result in errors in the line densities. For example, a MB of 0.1 ppm within a polygon would overestimate the line density by about $30\,\mathrm{kg\,m^{-1}}$ for a typical plume width of 20 km. For Berlin, we expect line densities of about 110 to $320\,\mathrm{kg\,m^{-1}}$ for emissions of $20\,\mathrm{Mt\,yr^{-1}}$, if we use wind speeds of 6 and $2\,\mathrm{m\,s^{-1}}$, respectively. As a result, the comparative

10   small bias of 0.1 ppm would result in an error in the estimated emissions of 10 and 30%. Since the patterns are rather random, the resulting errors would mostly show up in the SD of the background error and they would decrease if several line densities





were computed per plume. As plumes detected from $NO_2$ observations are larger, SDs are expected to be smaller. Indeed, this can be seen in the SD of the background error, which are about 50% versus 40% for $CO_2$- and $NO_2$-based plume detection, respectively (Figs. 8c and 9c).

5      The $CO_2$ background concentrations obtained from the plumes detected from the $CO_2$ observations are slightly higher (0.08 ppm on average) than the backgrounds from the plumes detected from the $NO_2$ observations. The reason is that the plumes detected from the $CO_2$ observations are smaller and thus pixels outside the plume have higher $CO_2$ values, because they still contain some enhanced values from the Berlin plume or other smaller plumes in the vicinity. As a consequence, the size of the detected plume has an impact on the background error. For $CO_2$-based plume detection, the emissions are underestimated by 10 to 30%. The effect increases with instrument noise, because detected plume size decreases.

10      On the other hand, the plume detected from the $NO_2$ observations is larger and thus includes all emission of Berlin but might also include emissions in the vicinity of Berlin. Therefore, the mass-balance approach does not only estimate emissions from Berlin, but also emissions from sources outside. Since there are no large point sources just outside the city boundaries, emissions from outside of Berlin are relative small. The $CO_2$ emissions from outside Berlin are $4.2\,\mathrm{Mt\,yr^{-1}}$ (20% of Berlin's annual emissions at overpass) within a radius of 25 km around Berlin's city center. On average, emissions estimated from the plume detected from the $NO_2$ observations are about 10% higher than Berlin's emissions. It is therefore necessary to interpret the emissions determined with the mass balance approach as emissions from a footprint that may be larger than the area of the city (e.g., Pitt et al., 2019).

### 4.3.4    Wind error

The wind error computed here includes only the difference between the mean wind below 500 m and the effective mean wind speed within the detected plume. For the $NO_2$-based plume detection, the mean wind error is close to zero ($+0.04\,\mathrm{m\,s^{-1}}$, i.e. <1%) with a SD of about $1.6\,\mathrm{m\,s^{-1}}$ (32%) for the 34 cases with successful estimates. The mean effective wind speed for these cases is $5.2\,\mathrm{m\,s^{-1}}$ that has been used to compute the relative error. If $CO_2$ observations are used for detecting the plume, the mean difference increases from 0.5 to $0.7\,\mathrm{m\,s^{-1}}$ (14-19%) for the low to high-noise scenario and the SD is about $1.2\,\mathrm{m\,s^{-1}}$ (33%) where the effective wind speed was $3.7\,\mathrm{m\,s^{-1}}$ for the 16 plumes.

25      The small MB with $NO_2$-based plume detection shows that the mean wind between 0 and 500 m was a suitable estimate of the effective wind speed in the detected plume. The MB increases for the $CO_2$-based plume detection, because mostly pixels in the vicinity of Berlin are detected from the $CO_2$ observations, while the plumes detected from the $NO_2$ observations extend further downstream. In the vicinity of the sources, the effective plume height is lower than further downstream, because the city plume has undergone less vertical mixing. Consequently, the effective wind speed is also smaller, because wind speed is lower near the surface. The small overestimation of the wind speed results in a significant overestimation of emissions of about 6% and 14 to 22% for the $NO_2$- and $CO_2$-based plume detection, respectively. The SD of the wind error leads to an error in the estimated emissions of 30 to 40%.

     We used the wind profile over the city of Berlin instead of the wind inside the plume to account for model errors in the estimated wind speed. As a result, SDs of the wind error were quite large ($1\text{-}2\,\mathrm{m\,s^{-1}}$) but likely realistic, because they are of a





similar magnitude as difference between measured and simulated winds in model validation studies (e.g., Sharp et al., 2015). Although the mean wind between 0 and 500 m was found to be a suitable estimate of the effective wind speed in this study, the choice of altitude range was rather arbitrary. Averaging the wind, for example, between 0 and 1 km would overestimate emissions by 15% on average as the wind speeds are higher. Choosing an optimal altitude range is thus one of the largest
challenges of the mass balance approach.

### 4.3.5   Total error

The breakdown of the errors shows that method, background, and wind error strongly contribute to the total error, while the influence of the retrieval error is comparatively small. Since the background error is negative for plumes detected from the $CO_2$ observations, the positive retrieval and wind errors are partially compensated resulting in the decrease of the MB with
increasing instrument noise (Table 3).

Our study did not include systematic retrieval errors from aerosols, clouds, and surface reflectance. Systematic errors can lead to large-scale biases, which would not affect the results if they influenced the observations inside and outside the city plume in the same way. However, systematic error patterns correlated with the $CO_2$ plumes, caused for example by enhanced aerosol concentrations in the city plume, could lead to biased emission estimates. Such effects are currently investigated in
a study on the use of aerosol information for estimating fossil fuel $CO_2$ emissions (AEROCARB). They showed that the proposed CO2M aerosol instrument (i.e. a multi-angle polarimeter) can reduce systematic errors due to aerosols to a level suitable for monitoring $CO_2$ emissions from cities (Houweling et al., 2019).

Overall, total errors were smallest when the $NO_2$ observations were available for detecting the plume and constraining the width of the transect. The errors presented here could likely be reduced further by better accounting for the temporal variability
of emissions, the variability of wind speed within the plume, and more generally by incorporating any other information from models or observations that helps to constrain the approach.

### 4.4   Estimating annual emissions

The results up to now focused on estimated emissions at individual satellite overpasses. To obtain annual emissions (at overpass time), seasonal cycles were fitted to the individual estimates of the analytical inversion and the mass balance approach. For the
analytical inversion, the uncertainty of the individual estimates were taken from the uncertainties obtained from the algorithm. For the mass-balance approach, we used an uncertainty of $10\,\mathrm{Mt\,yr^{-1}}$ (50% of emissions at overpass) based on the estimated uncertainties in the approach. The results are shown only for the medium noise scenario and for the method where line densities were computed from the mean values in the sub-polygons.

Figure 2a and c show the seasonal cycle fitted for a constellation of three satellites to the emission estimated by the analytical
inversion. For the time-constant emissions, the annual emissions obtained from the fitted seasonal cycle ($17.5\pm0.4\,\mathrm{Mt\,yr^{-1}}$) agree well with the true annual emissions ($16.8\,\mathrm{Mt\,yr^{-1}}$). For the time-varying emissions, the seasonal cycle is also fitted well but emissions are slightly overestimated in winter where no emission estimates are available. As a result, the estimated annual emissions of $21.6\pm0.4\,\mathrm{Mt\,yr^{-1}}$ are slightly larger than the true emissions at overpass ($20.0\,\mathrm{Mt\,yr^{-1}}$).





Since the number of estimates is lower with the mass-balance approach, fitting the seasonal cycle is more challenging, in particular, for the few estimates from the $CO_2$ observations alone (Figure 7a). Nonetheless, annual emissions were estimated quite well with $20.9\pm3.8\,\mathrm{Mt\,yr^{-1}}$ for a constellation of three satellites. If $NO_2$ observations are used for detecting the plumes, the temporal coverage is better and the seasonal cycle is fitted better, but emissions are overestimated in early summer. As a

result, annual emissions are also higher with $22.3\pm2.5\,\mathrm{Mt\,yr^{-1}}$.

To analyse the effect of constellation size, we estimated annual emissions for constellations of 1, 2, 3 and 6 satellites (Figure 11). Under the assumption of a perfect model, the analytical inversion is able to estimate annual emissions well for both constant and time-varying emissions even with only one satellite (panels a and b) with an average precision of $1.1\,\mathrm{Mt\,yr^{-1}}$ (5.5%). The time-varying emissions are slightly overestimated, because the fitted seasonal cycle tends to overestimate emissions in winter

due to missing satellite overpasses in these months.

In contrast, estimating annual emissions with the mass-balance is very difficult, if only the $CO_2$ observations are available for detecting the location of the plume. For a single satellite, the number of overpasses with successful emission estimates is too small to fit a seasonal cycle in nearly all cases and even with two or three satellites the precision is low with about $8.8\,\mathrm{Mt\,yr^{-1}}$ (44%). The situation significantly improves with additional $NO_2$ observations. A single satellite can estimate annual emissions

in five out of six cases with an average precision of $5.1\,\mathrm{Mt\,yr^{-1}}$ (26%) due to the better temporal coverage. The average precision increases to $4.4\,\mathrm{Mt\,yr^{-1}}$ (22%) and $2.5\,\mathrm{Mt\,yr^{-1}}$ (13%) for constellations of two and three satellites, respectively. The annual emissions are slightly overestimated, because of low temporal coverage in winter and because the mass-balance approach is also sensitive to emissions outside of Berlin.

To estimate annual emissions accurately, it is necessary to resolve the real temporal variability of emissions. It should be

noted that the temporal variability in the COSMO-GHG simulations used for generating the synthetic satellite observations likely underestimates the real variability. To generate temporally varying emissions in the simulations, we applied different diurnal, weekly and seasonal cycles to emissions from different sectors such as energy production, traffic and heating (Jähn et al., 2020). These fixed time profiles do not account for effects from meteorology and human drivers such as strikes, temporal traffic restrictions or holidays.

To resolve the temporal variability, a sufficiently large number of individual emission estimates is required. The number of estimates varies strongly between satellites due to the uneven distribution of cloud-free days. Even with the $NO_2$ observations, it is still possible to have only four overpasses with estimated emissions per year with two satellites. Therefore, at least three satellites are likely necessary to get reliable estimates of the annual emissions. Higher temporal coverage can alternatively be achieved by increasing the swath width of the instrument.

It should be noted that emissions at overpass cannot directly be compared to true annual emissions since the individual estimates are only representative of emissions a few hours before the satellite overpass (Broquet et al., 2018) but not for the daily mean. It is therefore still necessary to apply a correction factor to obtain the annual mean emissions, which introduces an additional source of uncertainty not included in our estimate.

Furthermore, it should be noted that our results are representative for a city in mid-latitudes, where the temporal coverage

is larger, as the satellites can pass over a city twice during the 11-day repeat cycle, whereas at the equator only one satellite



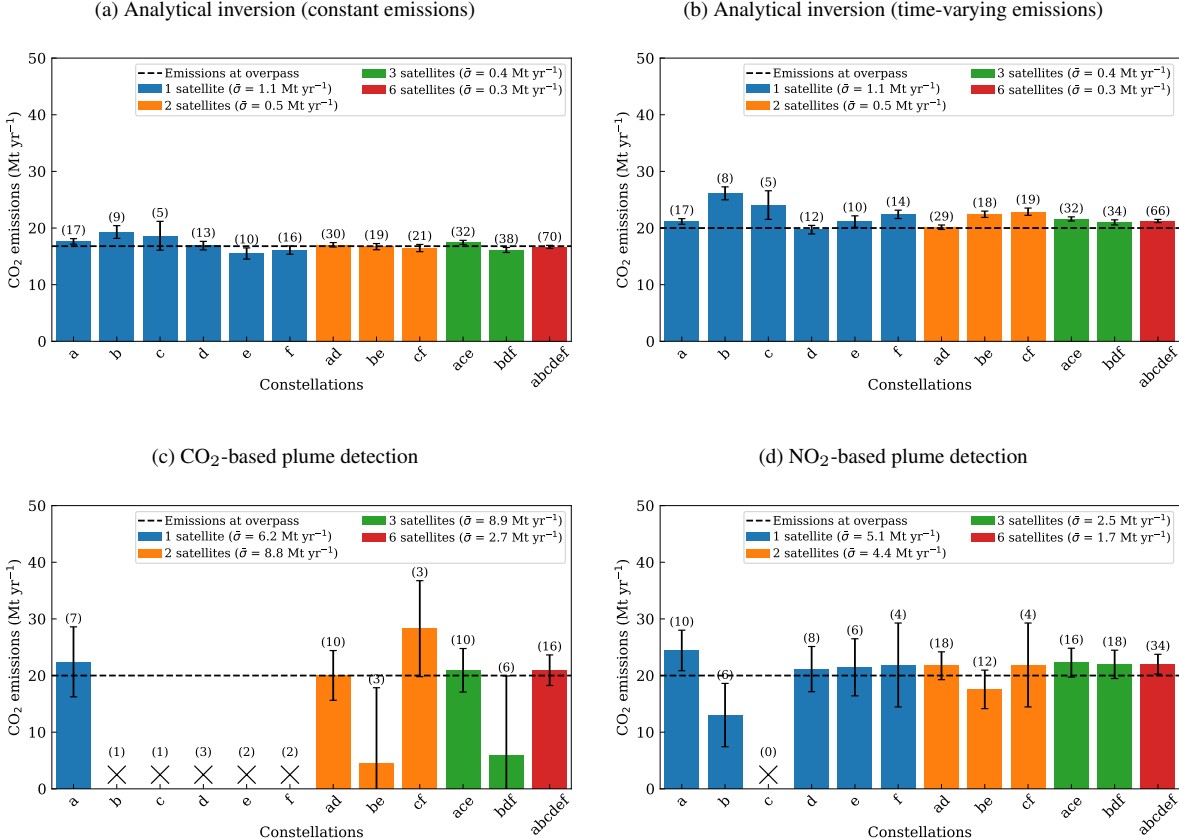

**Figure 11.** Estimated annual emissions at satellite overpass for (a) analytical inversion, (b) $CO_2$- and (c) $NO_2$-based plume detection for different constellation sizes. The number of overpasses with estimated emissions are shown in brackets. If the number of estimated emissions is too small for computing the seasonal cycle values are marked with a cross. Error bars show the precision computed from the individual emission estimates at satellite overpass. The legend shows the mean error for each constellation size.

pass takes place per 11 days. The real temporal coverage is also strongly affected by the number of cloud free observations in different latitudes.

## 5 Discussion and conclusions

In this study, a detailed analysis was conducted to investigate the potential of a constellation of $CO_2$ satellites with imaging
5 capability for quantifying the emissions of a large city like Berlin with or without additional $NO_2$ measurements. The results are based on unique, one year long very high resolution (1 km × 1 km) atmospheric $CO_2$ simulations with the COSMO-GHG model, which accounts for anthropogenic and biospheric fluxes as realistically as possible. Synthetic satellite observations





were generated for 2 km × 2 km pixels from $CO_2$ and $NO_2$ model fields along the 250-km wide swaths of a constellation of up to six satellites.

The $CO_2$ emissions of Berlin were quantified by two different methods to assess the range of uncertainties associated with different assumptions regarding the capabilities of atmospheric transport models. The emissions were quantified (1) by scaling

the simulated $CO_2$ tracer representing only emissions from Berlin to match the synthetic $CO_2$ observations without the $CO_2$ background field, and (2) by applying a mass-balance approach that estimates the flux of $CO_2$ through vertical control surfaces perpendicular to the direction of propagation of the detected plume. The second approach relies on a plume detection algorithm using either the $CO_2$ or $NO_2$ observations.

The first method assumes perfect knowledge of atmospheric transport and $CO_2$ background fields. In this case, the uncer-

tainty of the emission estimates is entirely driven by the ratio of instrument noise to the $CO_2$ enhancements within the plume, which varies from case to case due to varying winds and cloud cover. The second method requires minimal model information except for an estimate of the mean wind speed within the plume, which would typically be obtained from a numerical weather prediction model analysis.

The analytical inversion estimated emissions with a SD of 3.0 to $4.2\,\mathrm{Mt\,yr^{-1}}$ for the low to high-noise scenario and without

a bias because systematic retrieval errors were not included here. The average number of successful estimates is 11.0 per satellite and year (range: 5-17). For the mass-balance approach 2.7 plumes per satellite (range: 1-7) were available on average with $CO_2$-based plume detection and 5.7 plumes (range: 0-10) with the $NO_2$-based plume detection due to the better signal-to-noise ratio of the $NO_2$ observations. The mass-balance approach had a precision of about $10\,\mathrm{Mt\,yr^{-1}}$ for both $CO_2$- and $NO_2$-based plume detection.

The results obtained here can be compared with the Report for Mission Selection for CarbonSat that formulated a requirement of $7\,\mathrm{Mt\,yr^{-1}}$ uncertainty for single overpasses over a city with more than $35\,\mathrm{Mt\,yr^{-1}}$ ESA (2015). In our study, annual mean emissions of Berlin were much smaller ($16.8\,\mathrm{Mt\,yr^{-1}}$) than assumed in previous studies due to the use of a dedicated inventory provided by the city of Berlin. Since emissions are higher during daytime than during nighttime, the emissions at satellite overpass time (11:30 local time) are $20.0\,\mathrm{Mt\,yr^{-1}}$, which is still significantly smaller than $35\,\mathrm{Mt\,yr^{-1}}$. For this mag-

nitude of emissions the requirement of an uncertainty of $7\,\mathrm{Mt\,yr^{-1}}$ for single overpasses was clearly met under the assumption of a perfect model. When using a mass balance approach applied to the detected plumes, the requirement was almost met, irrespective of the uncertainty scenario used for the $CO_2$ instrument.

The emissions estimated with the mass-balance approach can have significant systematic errors due to the challenge of estimating the $CO_2$ background and the wind field accurately. Since the $NO_2$ observations make it possible to not only detect

the full city plume but also other small plumes in the vicinity, it is very helpful for estimating the $CO_2$ background field more accurately and also for filtering out scenes with interfering plumes from other sources. Additional $NO_2$ measurements on the same platform as the $CO_2$ measurements are thus highly beneficial not only for detecting the plume but also for estimating the $CO_2$ emissions. Furthermore, the analysis showed that $CO_2$ emission estimates do not depend strongly on the precision of the $CO_2$ observations for large plumes, for example, a city plume with more than 100 pixels. For these cases, a wider swath

and somewhat reduced $CO_2$ single sounding precision might be a reasonable trade-off. For smaller plumes, e.g., from power





plants, a high precision of the $CO_2$ observations is likely more relevant because of the small number of pixels contained in the plume.

Annual emissions were estimated by fitting a seasonal cycle to the individual estimates. The analytical inversion was able to estimate annual emissions with good precision with $1.1\,\mathrm{Mt\,yr^{-1}}$ (<6%) even with only one satellite, but this assumes perfect
knowledge of the atmospheric transport. Estimating the annual emissions was more challenging with the mass balance approach. If only the $CO_2$ measurements were available for estimating emissions, one satellite was not sufficient for estimating annual emissions in most cases, because the number of individual estimates was too small. The precision was still low with two or three satellites ($9\,\mathrm{Mt\,yr^{-1}}$ (44%)).

If $NO_2$ observations were available to detect the $CO_2$ plumes, the annual emission could be estimated with one satellite
in most cases (26% precision) due to the better temporal coverage. The precision improved further to $4.4\,\mathrm{Mt\,yr^{-1}}$ (22%) and $2.5\,\mathrm{Mt\,yr^{-1}}$ (13%) with two and three satellites, respectively. It should be noted that the uncertainty in an annual mean estimate derived from satellite observations does not only depend on the number of individual plume estimates but also on the magnitude and correlation structure of the temporal variability of the emissions. Therefore, it is necessary to study how many individual emission estimates are required to constrain this variability assuming realistic temporal correlations.

Our study suggests that the CO2M mission will be able to quantify annual emissions of a city like Berlin with high precision, even without knowledge about plume location and $CO_2$ background from a transport model, if additional $NO_2$ observations are available for detecting the plume and if the number of satellites is sufficiently large. With a population of 3.5 million, Berlin belongs to the 150 largest cities worldwide with more than 3 million inhabitants. The total population of these cities is 1.1 billion, which is roughly 15% of the world's population (United Nations, 2018). According to the analysis of $CO_2$ emission
clusters by Wang et al. (2019), there are also about 150 urban areas worldwide that have similar or higher $CO_2$ emissions than Berlin. Wang et al. (2020) showed that it might even be possible to constrain emissions of urban areas with emissions larger than $8\,\mathrm{Mt\,yr^{-1}}$, which would be about 300 cities.

Combining the mass-balance approach with additional information from models and other observations could further improve the accuracy of the $CO_2$ emission estimates if they would help to constrain critical aspects of the method such as the
position of the plume or the wind speed. The European $CO_2$ emission monitoring and verification support system, as envisioned to be implemented in the Copernicus program, would use the CO2M observations together with information from atmospheric transport models. Since spatial mismatches between real and simulated plumes may lead to large errors in the emission estimates, the system will have to account for uncertainties in simulated atmospheric transport. One way forward could thus be to develop an advanced data assimilation system able to extract wind information directly from the plume obser-
vation as demonstrated, for example, by Allen et al. (2013) for a 4D-Var ozone assimilation system. Since the shape and extent of the plume can be imaged more accurately from the $NO_2$ observations, the $NO_2$ measurements could be a very useful source of information in such a data assimilation system.



*Code and data availability.* Column averaged dry air mole fractions of all simulated tracers are available both as 2D fields and as synthetic satellite products through ESA.

*Author contributions.* GK developed, implemented, applied and evaluated the methods for estimating $CO_2$ emissions, and wrote the manuscript with input from all co-authors. DB supervised and led the project SMARTCARB. GB followed the project as external advisor and contributed

5    critical input to the manuscript. YM accompanied the study as ESA project officer and provided critical inputs and reviews during all phases of the project.

*Competing interests.* The authors declare that they have no conflict of interest.

*Acknowledgements.* We acknowledge funding of the project SMARTCARB by the European Space Agency (ESA) under contract no. 4000119599/16/NL/FF/mg and support by the EU Horizon-2020 project CHE under grant no. 776186. The views expressed here can in

10    no way be taken to reflect the official opinion of ESA. The work was supported by a grant from the Swiss National Supercomputing Centre (CSCS) under project ID d73.



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
