# Peer review of "Quantifying CO2 emissions of a city with the Copernicus Anthropogenic CO2 Monitoring satellite mission"

_Atmospheric Measurement Techniques, 2020_

## Referee Comment (RC1) · Anonymous Referee #1 · 29 Jul 2020

The manuscript by Kuhlmann et al., entitled "Quantifying CO2 emissions of a city with the Copernicus anthropogenic CO2 monitoring satellite mission", attempts to evaluate the potential of a future satellite mission to constrain CO2 emissions from a city. I have major questions about the two methods used to estimate emissions and therefore propose the paper be reconsidered after major revisions.

The two methods included in the paper for emission estimation are an analytical inversion method and a mass-balance approach. The two methods are set up as a dichotomy and characterized as "encompassing the range between optimistic and pessimistic assumptions regarding the capability of atmospheric transport models". How-

ever, the optimistic nature of the inversion method is highly contrived. For instance, the inversion assumes perfect knowledge of atmospheric transport and the background $CO_2$ field (Eq. 1). Why do the authors not introduce transport errors of difference magnitudes to the inversion and examine the impacts on the retrieved emissions? Furthermore, estimation of the background $CO_2$ is a key methodological challenge in top-down urban emission quantification. The "y_BG" field used in Eq. 1 is a simulated quantity that cannot be observed in the real-world. Assuming y_BG is the "true" background $CO_2$ field, how, then, would the atmospheric inversion attempt to derive this background $CO_2$, and how would the resulting background error affect the atmospheric inversion? I would like to see both the transport and background error analyses incorporated into the atmospheric inversion.

The second, mass-balance method is referred to as "pessimistic". But I believe that it can also be viewed as optimistic since the assumed transport appears to be highly simplified, assuming simple averaging of wind vectors. However, my main criticism of the mass-balance method as it is presented the lack of clarity in its description. After re-reading Sect. 3.2 multiple times, I still have problems visualizing and understanding the methodology adopted by the authors. For instance, what does the "vertical control surface" look like? How is Fig. 1 relevant for the methodology? Probably what is lacking is a cartoon (perhaps in 3D) that describes the method visually, to help the reader grasp the relevant variables and geometry of the method. Also, I suggest that the authors consider using an actual case with a field of satellite-"observed" XCO2 such as shown in Fig. 4a within a figure like Fig. 1. Otherwise Fig. 1 is too abstract and removed from its actual application.

Finally, the authors did not consider diurnal variations in $CO_2$ emissions; just the seasonal pattern. Given the 11:30LT satellite overpass and systematic sampling of midday $CO_2$ distributions, what would be the potential sampling bias? This seems like a major omission. I strongly urge the authors include the diurnal sampling bias into the analyses.

With the aforementioned issues with the methodology, I reserve evaluation of the results and conclusions for a revised manuscript, when the issues have been addressed.

OTHER POINTS: Page 9: The "CO2-weighted wind vector" is used in the mass balance method. Shouldn't the wind vector also be weighted by air density as well and not just CO2 concentrations?

Figure 2b/d: What is meant by the numbers at the top of the panel: e.g., "60/74", "60/70", "59/73"? Should be explained in the figure caption.

Page 12: "rare opportunities for observing plumes in winter". Should explain why the wintertime opportunities are rare. Is this because of the solar zenith angle and ground snow cover?

Fig. 4a: A better color scale is needed. The plumes in Berlin are hardly visible. Another colorscale to be considered is to range from dark blue (negative) to dark red (positive), with gray in the middle, like in Fig. 10c. Also, the pixels appear quite noisy, with sign reversals from pixel to pixel. Why?

Fig. 4b: Is this the emissions represented on the y-axis from the entire city of Berlin? Clarify

---

## Referee Comment (RC2) · Anonymous Referee #2 · 4 Sep 2020

The paper describes simulation studies of CO2 inversions from the future CO2M satellite mission. It is shown how the number of satellites, instrument errors and errors in the methodology contribute to the overall CO2 inversion errors. Different configurations are compared. The paper is well written, and figures and tables are clear. The paper is very important material for the CO2M mission.

General remarks The paper uses the mean bias (MB) and standard deviation (SD) as metrics. However, in several cases also the term "error" is used. It is unclear what is meant by "error". I propose to only use the chosen metrics (in this cases MB and SD) and avoiding other terms.

[Figure]

**AMTD**

The SD is strongly influenced by outliers. Also, it assumes that errors are Gaussian distributed. I suggest that the authors also look into the use of other metrics, e.g. based on percentiles (for example 25% to 75% percentile range, or 16% to 50% and 50% to 18% if you want to compare to the SD). How would this change the presented results?

Code and data availability. "Column averaged dry air mole fractions of all simulated tracers are available both as 2D fields and as synthetic satellite products through ESA.". Given the importance of the study for the CO2M mission, all data should be made available online (by direct link) instead of upon request. Also, the authors should consider making the code available.

How are city emissions currently computed using ground based data and energy consumption data, and how do uncertainties therein compare with reported ones for CO2M?

Specific Comments

Section 2 The NOx chemistry is simplified to a large extent by using a constant lifetime and parametrized NO/NO2 ratio. How will these simplifications affect the end result?

Section 3.1 The matrix H provides the translation from the emission to the XCO2 field at any point in the domain. I believe that this is time dependent, i.e. information on atmospheric transport is contained in this matrix. Please include more information on the matrix H in this section.

"The plume may also contain emissions emitted earlier in the day, but this information is not available from the model". Why is this information not available?

Page 9, line 28 to 30 "To estimate the uncertainty . . . city plume". This procedure is not clear to me. Can you elaborate on the procedure to compute these effective winds?

Page 9: do you compute the fluxes for CO2 only, or also for NO2?

Caption Figure 3. What does the line represent? Please add this to the caption.

[Figure]

Figures 4 c-e and 5 c-e. The legend overlaps the data points, which is rather annoying. Please resolve this.

Table 3: I assume that the Median plume size is measured in number of ground pixels? Please add to the table and/or caption.

Line 32, p21: "In contrast, SDs are reduced when the curve is fitted by constraining its width using the NO2 observations resulting in the lowest retrieval errors." It is true that the SDs are the smallest, but not true for the MB. Therefore, I don't think that the retrieval errors are the lowest. Also, how do the authors define "retrieval errors"?

Section 4.3.2: it is stated that the retrieval error is mostly related to the instrument noise. The SD therefore increases with increasing noise. However, also the MB increases significantly with the instrument noise, contributing significantly to the total MB. I have two questions about that: 1. Why would the MB increase with the noise? 2. Can the positive MB be explained?

P 27, line 25 "the requirement of an uncertainty of 7 Mt yr−1 for single overpasses was clearly met under the assumption of a perfect model". This is true for the current setup, however certain important satellite retrieval errors were ignored. This shall be mentioned and discussed.
* * *

---

## Author Comment (AC1) · 27 Sep 2020

Please find our response in the attached supplement.

Please also note the supplement to this comment:
https://amt.copernicus.org/preprints/amt-2020-162/amt-2020-162-AC1-supplement.pdf

---

## Author Response (AR1)

**Quantifying $CO_2$ emissions of a city with the Copernicus Anthropogenic $CO_2$ Monitoring satellite mission**

Gerrit Kuhlmann et al.

**Response to the Reviewer's Comments**

We thank the two reviewers for their critical assessment and useful comments to improve the quality of our paper. In the following we address their concerns point by point.
* * *
**Reviewer 1**

5 **General comments**

**Reviewer Point P 1.1** — The two methods included in the paper for emission estimation are an analytical inversion method and a mass-balance approach. The two methods are set up as adichotomy and characterized as "encompassing the range between optimistic and pessimistic assumptions regarding the capability of atmospheric transport models". However, the optimistic nature of the inversion method is highly contrived. For instance, the
10 inversion assumes perfect knowledge of atmospheric transport and the background CO2 field (Eq. 1). Why do the authors not introduce transport errors of difference magnitudes to the inversion and examine the impacts on the retrieved emissions?

Furthermore, estimation of the background CO2 is a key methodological challenge in top-down urban emission quantification. The "y_BG" field used in Eq. 1 is a simulated quantity that cannot be observed in the real-world.
15 Assuming y_BG is the "true"background CO2 field, how, then, would the atmospheric inversion attempt to derive this background CO2, and how would the resulting background error affect the atmospheric inversion? I would like to see both the transport and background error analyses incorporated into the atmospheric inversion.

**Reply**: We agree that transport and background errors are important sources of uncertainty for quantifying $CO_2$ emissions from satellite observations. Transport model uncertainties are difficult to quantify, and they depend not only on the model
20 system but also on the meteorological observations available for data assimilation. One might argue that such uncertainties can be minimized in the future through further improvements in the model systems and the observation networks (Deng et al., 2017). Assessing the effects of transport model uncertainties on $CO_2$ concentrations has been attempted previously using ensemble simulations (Lauvaux et al., 2009; McNorton et al., 2020). Such simulations are computationally very

expensive and were, therefore, not feasible in our case, where already a single simulation over a 1 year period at 1 km x 1 km resolution was very costly. Note that it is not at all uncommon to assume perfectly known transport in Observing System Simulation Experiments for urban CO2 emissions (Pillai et al., 2016; Broquet et al., 2018).

An inversion system can estimate the $CO_2$ background by replacing scalar $x$ (Eq. 1) with a vector where a scaling factor is added for the simulated background tracer $y_{bg}$. A more complex system could optimize various components of the background (e.g. anthropogenic and biospheric fluxes) independently. To access the effect of background error on the inversion, a model ensemble using, for example, different emissions inventory is needed (e.g. Pillai et al., 2016), which would also require additional and costly simulations.

In our study design we have, therefore, chosen a different approach: We have applied two independent and highly complementary methods, the first one assuming that atmospheric transport and background field are known perfectly, the second one, in contrast, assuming that essentially all information has to be retrieved from the observations alone. The first approach allows us to study the effects of satellite measurement uncertainties and the influence of the size of the constellation irrespective of a specific transport model. This represents the most optimistic case where everything is perfectly known except for the amplitude of the emissions from the city. The second approach allows us to determine how well emissions can be quantified under the pessimistic assumption, that atmospheric transport models cannot provide any useful information linking atmospheric $CO_2$ concentrations to emissions from a city, but that this information has to be determined from the satellite observations alone. Our mass-balance approach also makes the assumption that the $CO_2$ background is a smooth field that has to be determined from satellite pixels outside the city plume.

We believe that this is a valid approach: Our analytical inversion is somewhat simpler than than the inversion methods applied in Pillai et al. (2016) and Broquet et al. (2018), where also uncertainties in prior fluxes and background were considered (but no transport uncertainties), but it goes well beyond those studies by including a second, entirely data-driven approach.

We have updated the abstract and the following paragraph in the introduction to better describe our approach:

> The first approach assumes that the atmospheric transport is known perfectly. It uses an analytical inversion that is applied to the simulated plume signature of the city provided by the same model used to generate the synthetic observations. This approach follows the general concept used in previous OSSEs studies (Pillai et al., 2016; Broquet et al., 2018). It does not account for the effect of model errors on the estimated emissions, in particular, the challenge to correctly simulate the location of the emissions plume, which was also not considered in previous studies. We also assume that the $CO_2$ background field from anthropogenic and natural fluxes outside the city can be obtained appropriately from the simulations.

**Reviewer Point P 1.2** — The second, mass-balance method is referred to as "pessimistic". But I believe that it can also be viewed as optimistic since the assumed transport appears to be highly simplified, assuming simple averaging of wind vectors.

**Reply**: We respectfully disagree: Approximating the complex atmospheric flow that is present in the simulations (mimicking reality) by a simple mean wind speed is a pessimistic not an optimistic assumption, since it assumes that an atmospheric transport model cannot provide any useful information on the flow within the plume except for an (uncertain) estimate of the mean wind speed.

5    **Reviewer Point P 1.3** — However, my main criticism of the mass-balance method as it is presented the lack of clarity in its description. After re-reading Sect. 3.2 multiple times, I still have problems visualizing and understanding the methodology adopted by the authors. For instance, what does the "vertical control surface" look like? How is Fig. 1 relevant for the methodology? Probably what is lacking is a cartoon (perhaps in 3D) that describes the method visually, to help the reader grasp the relevant variables and geometry of the method. Also, I suggest that the authors
10    consider using an actual case with a field of satellite-"observed" XCO2 such as shown in Fig. 4a within a figure like Fig. 1. Otherwise Fig. 1 is too abstract and removed from its actual application.

**Reply**: We agree that the method was not sufficiently well explained. We have revised Section 3.2 (see diff file for details), removed the term "vertical control surface", and replaced Figure 1 with the following, more detailed schematic:

[Figure]

**Figure 1.** (a) Sketch of a $CO_2$ city plume with detected pixels and fitted center line. Random noise has been added to the $CO_2$ observations. The center of the city source is denoted by $S$. The origin of the center curve is $O = (x_o, y_o)$. For a satellite pixel $P$, the across-plume coordinate $y_p$ is the distance between $P$ and $Q$, and the along-plume coordinate $x_p$ is the arc length from $S$ to $Q$. The yellow rectangles are the polygons used for computing the line densities. (b) Example of $CO_2$ mass columns in across plume distance for the polygon containing the pixel $P$. (c) Line densities computed for each polygon in the sketch. The line densities are zero upstream of the source, build up over the city, and remain constant downstream of the city.

**Reviewer Point P 1.4** — Finally, the authors did not consider diurnal variations in CO2 emissions; just the
15    seasonal pattern. Given the 11:30 LT satellite overpass and systematic sampling of mid-day CO2 distributions, what would be the potential sampling bias? This seems like a major omission. I strongly urge the authors include the diurnal sampling bias into the analyses.

**Reply**: Diurnal variations of $CO_2$ emissions were accounted for in the simulations, but indeed, we did not attempt to provide an estimate of the sampling bias introduced by the fact that the satellite is most sensitive to the emissions a few hours prior to the overpass but not to the diurnal mean. We have included the following more detailed discussion in Section 4.4:

Besides day-to-day variability of emissions, individual emission estimates are also only representative of emissions a few hours before the satellite overpass but not for the daily mean (Broquet et al., 2018). It would therefore still be necessary to apply a correction factor to obtain the annual mean emissions, which introduces an additional source of uncertainty not included in our estimate. In our simulation, the emissions during overpass are about 18% higher than the daily mean, suggesting that the sampling bias would be of the same order of magnitude. However, this result is entirely driven by the sector-specific diurnal emission cycles prescribed in the simulations. If the diurnal cycle of emissions was known from other sources of information such as traffic counts, electricity demand and heating statistics, a correction of the sampling bias could be applied, but this correction would add an additional uncertainty. The uncertainty in current estimates of diurnal emission variations is very poorly known, which makes it difficult to derive uncertainties in diurnal cycle or precise knowledge about ratios between different periods of the day (Wang et al., 2020). Studies such as those of Nassar et al. (2013), Gurney et al. (2019), or Peylin et al. (2011) all present diurnal emission cycles from various sources of information, but no analysis of uncertainties. The diurnal cycles presented in these studies are roughly in line with our estimate of a sampling bias of the order of 20% with respect to the daily mean. Super et al. (2020) estimated uncertainties in the diurnal variation from uncertainties in activity data and emission factors, which, when applied to individual cities, would make it possible to better quantify the potential sampling bias in estimating annual emissions from sun-synchronous satellite observations in the future.

**Specific comments**

**Reviewer Point P 1.5** — Page 9: The "CO2-weighted wind vector" is used in the mass balance method. Shouldn't the wind vector also be weighted by air density as well and not just CO2 concentrations?

**Reply**: The effective wind speed accounts for the $CO_2$ mass in a model grid cell. We have revised the paragraph to better explain how the effective wind speed is computed:

To estimate the uncertainty in this simplified estimate of the mean flow speed, we also computed an effective wind speed for each polygon, taking into account the three-dimensional distribution of winds and $CO_2$ mass concentrations. The effective wind speed is the weighted mean wind speed parallel to the plume's center line and weighted by the partial $CO_2$ mass column density of the plume. These $CO_2$ column densities are taken from the model tracer that contains only emissions of the city.

**Reviewer Point P 1.6** — Figure 2b/d: What is meant by the numbers at the top of the panel: e.g., "60/74","60/70", "59/73"? Should be explained in the figure caption.

**Reply**: The figure caption has been updated as follows:

The numbers above the boxes are the cases where the uncertainties for all three scenarios are less than 50% (first number) and the number of successful emission estimates for each scenario (second number).

as well as the caption of Figure 7:

The numbers above the boxes are the number of cases for which emissions could be estimated for all three scenarios (first number) and the number of successful emission estimates for each scenario (second number).

**Reviewer Point P 1.7** — Page 12: "rare opportunities for observing plumes in winter". Should explain why the wintertime opportunities are rare. Is this because of the solar zenith angle and ground snow cover?

**Reply**: The main reason here was frequent cloud cover. We have updated the sentence:

The rare opportunities for observing plumes in winter due to frequent cloud cover, however, can easily be missed by a small constellation of satellites, which will make it difficult to reliably trace the seasonal cycle

**Reviewer Point P 1.8** — Fig. 4a: A better color scale is needed. The plumes in Berlin are hardly visible. Another colorscale to be considered is to range from dark blue (negative) to dark red (positive), with gray in the middle, like in Fig. 10c. Also, the pixels appear quite noisy, with sign reversals from pixel to pixel. Why?

**Reply**: The reason that Berlin's $CO_2$ plume is hardly visible is that its signal-to-noise ratio is close to 1. We have added the following sentence to Section 4.2.1:

In Figure 4a, the $CO_2$ plume is hardly visible, because its signal-to-noise ratio is close to 1. In contrast, the $NO_2$ plume is clearly visible in the $NO_2$ image.

We have also updated the range of the colorbar from [-2,+2] to [-1.5,1.5] around the mean to improve its visibility slightly. Since the issue is the small signal-to-noise ratio, changing the colormap will not increase the visibility of the plume.

We have also updated the colorbar labels, because Figure 4a did not show the absolute $XCO_2$ signal, but the $XCO_2$ field minus the mean $XCO_2$ values in the swath. For this reason, the pixels reversed sign. We have updated figure now showing absolute $XCO_2$ values [404-407 ppm] and have also replaced Figure 5a accordingly:

[Figure]

**Reviewer Point P 1.9** — Fig. 4b: Is this the emissions represented on the y-axis from the entire city of Berlin? Clarify

**Reply**: Yes, this is the flux of the entire city of Berlin. We have updated the caption to better describe Figure 4b

(b) $CO_2$ flux and wind speed as a function of along-plume distance. The $CO_2$ fluxes estimated from the line densities are shown as markers with their uncertainty for the noise-free model tracer and the synthetic satellite observations. The horizontal lines show true emissions of Berlin at 10:30 UTC (black line) and the estimated emissions. The extent of the city is highlighted by the light gray area.
* * *
**Reviewer 2**

**General comments**

**Reviewer Point P 2.1** — The paper uses the mean bias (MB) and standard deviation (SD) as metrics. However, in several cases also the term "error" is used. It is unclear what is meant by "error". I propose to only use the chosen metrics (in this cases MB and SD) and avoiding other terms.

**Reply**: We have revised and changed several instances of "error" to be more specific when referring to MBs and SDs of the different error types.

**Reviewer Point P 2.2** — The SD is strongly influenced by outliers. Also, it assumes that errors are Gaussian distributed. I suggest that the authors also look into the use of other metrics, e.g. based on percentiles (for example 25% to 75% percentile range, or 16% to 50% and 50% to 18% if you want to compare to the SD). How would this change the presented results?

**Reply**: We have computed the 16-84th percentile range (PR) for the different error types and divided this range by two to compare with SDs (see Table in Supplement). While SDs and PRs show some differences, the overall results in this study are not sensitive to this choice, suggesting that the SDs are not strongly affected by outliers.

**Reviewer Point P 2.3** — Code and data availability. "Column averaged dry air mole fractions of all simulated tracers are available both as 2D fields and as synthetic satellite products through ESA.". Given the importance of the study for the CO2M mission, all data should be made available online (by direct link) instead of upon request. Also, the authors should consider making the code available.

**Reply**: We will upload the data to a public data repository and provide the link with the accepted publication. We will publish our codes, i.e. plume detection and mass-balance approach, upon completion of the SMARTCARB project. We updated the code and data availability section accordingly:

> $XCO_2$, CO and $NO_2$ columns of all simulated tracers are available both as 2D fields and as synthetic satellite products (Kuhlmann et al. 2020). The code used in the publication is available on request and will be published on the group's Gitlab group (https://gitlab.com/empa503) upon completion of the SMARTCARB project.

**Reviewer Point P 2.4** — How are city emissions currently computed using ground based data and energy consumption data, and how do uncertainties therein compare with reported ones for CO2M?

**Reply**: We added the following sentences to the conclusions:

> Estimates of the city emissions are currently compiled in emission inventories based on activity data, energy statistics, emission factors and self-reported emissions. The uncertainties in total city emissions have a large range depending on data availability, and many cities do not even have an inventory (Gately and Hutyra, 2017). The characterization of uncertainties in inventories is a complex topic, since the characterization of uncertainties in the input parameters and thus the propagation of uncertainties is difficult (Super et al., 2020). The detailed emission inventory used for Berlin in our study reports only sector specific uncertainties from which we estimate that the uncertainty in total emissions to be around 25-30% (AVISO GmbH and IE Leipzig, 2016). Our study therefore suggests that the CO2M mission will be able to quantify annual emissions of a city like Berlin with higher precision, even without knowledge about plume location and $CO_2$ background from a transport model, if additional $NO_2$ observations are available for detecting the plume and if the number of satellites is sufficiently large.

Note that we have contacted the city of Berlin to discuss our estimate with them. Since our contact person is currently on leave, we may need to adjust this paragraph based on their response.

**Specific comments**

**Reviewer Point P 2.5** — Section 2 The NOx chemistry is simplified to a large extent by using a constant lifetime and parametrized NO/NO2 ratio. How will these simplifications affect the end result?

**Reply**: We have added the following paragraph to Section 4.3.5:

5       The $NO_x$ chemistry used in our simulations was highly simplified accounting only for a constant $NO_x$ lifetime of four hours. Laughner and Cohen (2019) recently estimated $NO_x$ lifetimes of North American cities from satellite observation and found annual mean lifetimes varying between 1 and 5 hours for different cities. In our study, a shorter lifetime would result in $NO_2$ signals that decrease faster downstream and therefore the detectable plume would be correspondingly shorter. A different plume length will

10       reduce the number of polygons available for computing line densities and thus could impact the SD of the retrieval error. Berlin's mean plume length was about 90 km for plumes detected from the $NO_2$ observations. A lifetime of 2 hours would reduce the plume length to about 45 km, which is the mean plume length from plumes detected with the low-noise $CO_2$ observations. The SDs of the retrieval error are very similar between the shorter and longer plumes detected from the $CO_2$ and $NO_2$ observations,

15       respectively (Table S1 and S2 in the Supplement). This suggests that the higher signals near the source are best for accurate estimation of line densities, while $CO_2$ observations further downstream do not improve the emission estimate, because the line densities estimated for these more diluted parts of the plume are more uncertain. We therefore expect that a shorter lifetime does not affect our results. A full-chemistry simulation would be necessary to fully understand the impact of $NO_x$ chemistry on the

20       mass balance approach.

In this context, we also added the following sentences to the last paragraph in Section 4.3.5.:

[...] It might be possible to reduce uncertainties when limiting the analysis to polygons near the source where the plume is not yet strongly diluted by turbulent mixing. The optimal distance presumably depends on wind speed and atmospheric stability and has not been analysed here. [...]

25 **Reviewer Point P 2.6** — Section 3.1 The matrix H provides the translation from the emission to the XCO2 field at any point in the domain. I believe that this is time dependent, i.e. information on atmospheric transport is contained in this matrix. Please include more information on the matrix H in this section.

**Reply**: The matrix is built with information from the atmospheric transport model. While the matrix is different for each satellite overpass, it does not contain information about the temporal variability of emissions. We have extended the

30 description as follows:

$\mathbf{H}$ is the observation operator representing the sensitivity of the $XCO_2$ signal to changes in $x$, i.e. the emissions, in the satellite image. Since $x$ is a scalar, $\mathbf{H}$ is a row matrix. It contains all $XCO_2$ values obtained from the $CO_2$ tracer simulated with constant emissions of Berlin that are larger than 0.05 ppm.

**Reviewer Point P 2.7** — "The plume may also contain emissions emitted earlier in the day, but this information is not available from the model".

Why is this information not available?

**Reply**: We have reformulated the sentence:

The plume may also contain emissions emitted earlier in the day, but the observation operator does not include information about the temporal variability of emissions.

**Reviewer Point P 2.8** — Page 9, line 28 to 30 "To estimate the uncertainty...city plume". This procedure is not clear to me. Can you elaborate on the procedure to compute these effective winds?

**Reply**: We have revised the sentence to better explain our procedure:

To estimate the uncertainty in this simplified estimate of the mean flow speed, we also computed an effective wind speed for each polygon, taking into account the three-dimensional distribution of winds and $CO_2$ mass concentrations. The effective wind speed is the weighted mean wind speed parallel to the plume's center line and weighted by the partial $CO_2$ mass column density of the plume. These $CO_2$ column densities are taken from the model tracer that contains only emissions of the city.

We also added the following sentence to Section 4.2.1:

The figures also show the mean and effective wind speed along the plume. While the average wind speed is constant, the effective wind speed is lower near the city center where the $CO_2$ plume is still near the surface where wind speeds are lower. The mean height of the plume increases downstream of the city and, therefore, the effective wind speed also generally increases with distance from the city.

**Reviewer Point P 2.9** — Page 9: do you compute the fluxes for CO2 only, or also for NO2?

**Reply**: We only compute $CO_2$ fluxes in this study.

**Reviewer Point P 2.10** — Caption Figure 3. What does the line represent? Please add this to the caption.

**Reply**: The lines are the fits of the multilinear regression model. We have updated the figure caption.

**Reviewer Point P 2.11** — Figures 4 c-e and 5 c-e. The legend overlaps the data points, which is rather annoying. Please resolve this.

**Reply**: We have updated these figure avoiding overlapping of legend and data points.

**Reviewer Point P 2.12** — Table 3: I assume that the Median plume size is measured in number of ground pixels? Please add to the table and/or caption.

**Reply**: Done.

**Reviewer Point P 2.13** — Line 32, p21: "In contrast, SDs are reduced when the curve is fitted by constraining its width using the NO2 observations resulting in the lowest retrieval errors." It is true that the SDs are the smallest, but not true for the MB. Therefore, I don't think that the retrieval errors are the lowest. Also, how do the authors define "retrieval errors"?

**Reply**: We changed "lowest retrieval errors" to "lowest SDs of the retrieval errors". Retrieval errors represents the impact of the random noise in the $CO_2$ observations on the calculation of line densities as described at the end of Section 3.2.

**Reviewer Point P 2.14** — Section 4.3.2: it is stated that the retrieval error is mostly related to the instrument noise. The SD therefore increases with increasing noise. However, also the MB increases significantly with the instrument noise, contributing significantly to the total MB. I have two questions about that: 1. Why would the MB increase with the noise? 2. Can the positive MB be explained?

**Reply**: We have used this comment to rethink our previous answer to these questions in Section 4.3.2. We think that most likely explanation for the positive MB is that the plume detection algorithm is more likely to detect pixels with $CO_2/NO_2$ observations with positive random errors, which result in a bias in line densities. We have revised Section 4.3.2 accordingly:

> The retrieval error has a small positive bias that scales with the absolute noise of the uncertainty scenarios. The reason that the MB scales with the absolute noise is that the same random errors, i.e. spatial noise pattern, were applied to the three uncertainty scenarios of a scene except for a scaling required to achieve the standard deviation of the error $\sigma_{VEG50}$. Since no systematic errors were applied to the satellite observations, we would expect that the MB of the retrieval error is closer to zero. A likely explanation for the positive MB is that the plume detection algorithm is more likely to detect $CO_2$ pixels that are positive outliers, i.e. $CO_2$ values that have a large positive random error. If these outliers are included when computing line densities, they result in a positive bias in the estimated emissions. This artifact also affects plumes detected from $NO_2$ observations, because the same noise pattern was applied to $CO_2$

and NO$_2$ observations. While this is partly an artifact from setting up the OSSE in order to allow for comparison between the different instrument scenarios, it might also appear in real observations.

**Reviewer Point P 2.15** — P 27, line 25 "the requirement of an uncertainty of 7 Mt yr1 for single overpasses was clearly met under the assumption of a perfect model". This is true for the current setup, however certain important satellite retrieval errors were ignored. This shall be mentioned and discussed.

**Reply**: We updated the paragraph as follows:

For this magnitude of emissions the requirement of an uncertainty of 7 Mt yr$^{-1}$ for single overpasses was clearly met under the assumption that the CO$_2$ signature of the city plume and CO$_2$ background field can be simulated perfectly.

---

## Author Response (AR2)

**Quantifying $CO_2$ emissions of a city with the Copernicus Anthropogenic $CO_2$ Monitoring satellite mission**

Gerrit Kuhlmann et al.

**Response to the Reviewer's Comments**

We like to thank the reviewer again for their critical assessment and useful comments to improve the quality of the paper.
* * *
**Reviewer 1**

5 **Reviewer Point P 1.1** — I appreciate the authors' efforts at clarification in the main text. The addition of the new Fig. 1 explaining the mass balance approach is very helpful in explaining the mass balance methodology to the reader.

As discussed in my previous review, the background/transport errors and diurnal sampling errors are significant sources of uncertainty in interpreting satellite data, and I see that the authors are in agreement as well. While I
10 accept that the quantitative analyses can be incorporated in future analyses (albeit somewhat of a lost opportunity for this paper), I think the statements about the accuracy need to be qualified accordingly. Caveats need be added to the abstract, especially the part that starts with "The analytical inversion was able to estimate annual emissions with an accuracy of . . . " Given the fact that key uncertainties were neglected, a clear qualifying statement needs to be incorporated that refers to the fact that background/transport errors and diurnal sampling bias would result
15 in additional uncertainties. Note that the diurnal sampling bias applies to both the analytical inversion and mass balance methods.

**Reply**: We have revised the abstract and added the requested clarifications:

Annual emissions were estimated by fitting a low-order periodic spline to the individual estimates to account for the *seasonal* variability of the emissions *but we did not account for the diurnal cycle of*
20 *emissions, which is an additional source of uncertainty that is difficult to characterize*. The analytical inversion was able to estimate annual emissions with an accuracy of $<1.1\,\mathrm{Mt\,yr^{-1}}$ ($<6\%$) even with only one satellite, *but this assumes perfect knowledge of plume location and* $CO_2$ *background. The accuracy was much smaller when applying the mass balance approach, which determines plume location and background directly from the satellite observations.* At least two satellites were necessary for the mass-balance

approach to have a sufficiently large number of estimates distributed over the year to robustly fit a spline, but even then the accuracy was low ($>8\,\mathrm{Mt\,yr^{-1}}$ ($>40\%$)) when using the $CO_2$ observations alone. When using the $NO_2$ observations to detect the plume, the accuracy could be greatly improved to 22% and 13% with two and three satellites, respectively.

5 **Reviewer Point P 1.2** — Secondly, Figure 1 needs to include the two-dimensional curve p(r). Otherwise it is not connected to equations 3 and 4.

**Reply**: We have updated Figure 1a accordingly:

[Figure]

**Figure 1.** (a) Sketch of a $CO_2$ city plume with detected pixels and fitted center line. Random noise has been added to the $CO_2$ observations. The center of the city source is denoted by $S$. The origin of the center curve is $O = (x_o, y_o)$. For a satellite pixel $P$, the across-plume coordinate $y_p$ is the distance between $P$ and $Q$, and the along-plume coordinate $x_p$ is the arc length from $S$ to $Q$. The yellow rectangles are the polygons used for computing the line densities. (b) Example of $CO_2$ mass columns in across plume distance for the polygon containing the pixel $P$. (c) Line densities computed for each polygon in the sketch. The line densities are zero upstream of the source, build up over the city, and remain constant downstream of the city.